# BiNoMaP: Learning Category-Level Bimanual Non-Prehensile Manipulation Primitives

## Abstract

Non-prehensile manipulation, encompassing ungraspable actions such as pushing, poking, and pivoting, represents a critical yet underexplored domain in robotics due to its contact-rich and analytically intractable nature. In this work, we revisit this problem from two novel perspectives. First, we move beyond the usual single-arm setup and the strong assumption of favorable external dexterity such as walls, ramps, or edges. Instead, we advocate a generalizable dual-arm configuration and establish a suite of Bimanual Non-prehensile Manipulation Primitives (BiNoMaP). Second, we depart from the prevailing RL-based paradigm and propose a three-stage, RL-free framework to learn non-prehensile skills. Specifically, we begin by extracting bimanual hand motion trajectories from video demonstrations. Due to visual inaccuracies and morphological gaps, these coarse trajectories are difficult to transfer directly to robotic end-effectors. To address this, we propose a geometry-aware post-optimization algorithm that refines raw motions into executable manipulation primitives that conform to specific motion patterns. Beyond instance-level reproduction, we further enable category-level generalization by parameterizing the learned primitives with object-relevant geometric attributes, particularly size, resulting in adaptable and general parameterized manipulation primitives. We validate BiNoMaP across a range of representative bimanual tasks and diverse object categories, demonstrating its effectiveness, efficiency, versatility, and superior generalization capability.

## 1 Introduction

Non-prehensile manipulation refers to a class of robotic actions that do not rely on firm grasping but instead leverage physical interactions such as poking, or pivoting, or pushing to achieve manipulation goals Zhou et al. (2019); Hogan & Rodriguez (2020); Sun et al. (2020); Zhou & Held (2023); Zhang et al. (2023). These skills are not merely complementary to traditional grasp-based tasks; they are often essential in scenarios where grasping is physically infeasible or inefficient. In dual-arm robotic systems Liu et al. (2022); Wu & Kruse (2024); Yamada et al. (2025); Lu et al. (2025), non-prehensile manipulation becomes especially relevant when dealing with objects that are too fragile, too flat, or lack sufficient geometry for reliable grasping.

Despite its importance, current non-prehensile manipulation faces two core bottlenecks. First, most existing works operate under the simplifying assumption of a unimanual setting, often coupled with highly structured environments Zhou et al. (2023); Cho et al. (2024); Wu et al. (2024); Lyu et al. (2025); Li et al. (2025a). To compensate for the lack of control authority, these methods rely on artificial aids such as vertical walls, inclined planes, or boundaries to stabilize and direct object motion. However, in real-world deployments, such assumptions are rarely satisfied due to environmental unpredictability or object fragility. Consider a scenario where a thin rectangular cardboard box lies flat on a tabletop—walls and ramps are unavailable, and poking the box may damage its contents. In such cases, a more general solution is to exploit bimanual coordination Krebs & Asfour (2022); Grannen et al. (2023), where one arm can serve as a stabilizing reference while the other executes the non-prehensile action. This configuration not only replaces inflexible external constraints with adaptive internal ones but also enables complex skills such as dual-arm wrapping Grotz et al. (2024); Lu et al. (2025); Zhou et al. (2025); Liu et al. (2025a) or cluttered object singulation Jiang et al. (2024b); Xu et al. (2025a), which are inaccessible to single-arm systems.

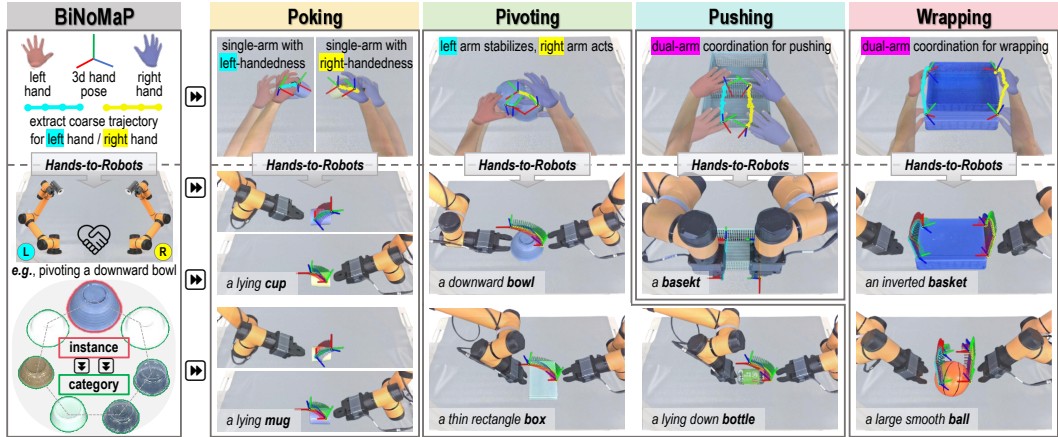

Figure 1: **Bi**manual **No**n-Prehensile **Ma**nipulation **P**rimitives (**BiNoMaP**). *(Left)* We propose to extract coarse hand trajectories of non-prehensile skills from human video demonstrations, and then refine and optimize them to the dual-arm robot. These reproduced skills can be further parameterized from instance-level to category-level. *(Right)* We extensively validated BiNoMaP on four skills (e.g., poking, pivoting, pushing, and wrapping) involving a variety of objects.

The second bottleneck lies in the heavy reliance on reinforcement learning (RL) frameworks Schulman et al. (2017); Haarnoja et al. (2018); Fujimoto et al. (2018). Most advanced approaches require constructing task-specific simulators that model manipulators, top-tables, and object dynamics, followed by lengthy policy training with dense environment interactions and carefully engineered reward functions. These RL pipelines are often sensitive to hyperparameter tuning and face substantial sim-to-real gaps due to inaccuracies in simulated physics, including mass distributions, contact dynamics, or friction coefficients. While recent works attempt to mitigate this gap via world models or differentiable simulators Lyu et al. (2025); Li et al. (2025a); Huang et al. (2025), they still inherit the inherent limitations of RL, including slow convergence and poor generalization.

To this end, we present a three-stage paradigm that combines hardware generality with algorithmic efficiency. We employ a dual-arm setup with parallel-jaw grippers, which not only supports unimanual non-prehensile skills but also enables more complex bimanual ones. In the ***first*** stage, inspired by prior work on learning from human demonstrations Grauman et al. (2022; 2024); Chen et al. (2025); Papagiannis et al. (2025), we extract primitive bimanual motion trajectories from human hand videos for task-specific non-prehensile behaviors. Unlike grasp-based tasks, where 1–3 cm errors in hand-object alignment may be tolerable, non-prehensile tasks are extremely sensitive to deviations: even 3–5 mm misalignment can lead to premature contact loss or over-compression, causing instability or failure. To address this, our ***second*** stage introduces a geometry-aware post-optimization algorithm that leverages object shape priors to refine these noisy trajectories into smooth, task-specific motion primitives. These refined trajectories, which we term **Bi**manual **No**n-Prehensile **Ma**nipulation **P**rimitives (**BiNoMaP**), exhibit high success rates and are robust to object pose variations. The concept of primitive skills shares a similar spirit with seminal works focusing on primitive movement Schaal (2006); Nah et al. (2024), or behaviors Hogan & Sternad (2012); Barreiros et al. (2025). In the ***third*** stage, we further extend these primitives to unseen objects within the same category by parameterizing them with object-specific geometric attributes, such as the length and width of a box or the diameter of a sphere. This results in a family of ***Parameterized Manipulation Primitives*** that are adaptive and transferable across diverse category-level instances.

We extensively evaluate BiNoMaP on a diverse set of non-prehensile dual-arm skills, including poking, pivoting, pushing, and wrapping (Fig. 1). We test them on various objects with varying shapes, materials, and physical properties. To prove the effectiveness and efficiency, we compare against strong baselines Zhao et al. (2023a); Chi et al. (2023); Ze et al. (2024); Zhou et al. (2023); Cho et al. (2024); Lyu et al. (2025). Our results show that BiNoMaP consistently achieves higher success rates across tasks. Furthermore, we demonstrate how BiNoMaP can be integrated with high-level vision-language models (VLMs) Xiao et al. (2024); Ravi et al. (2025) to support advanced robotic behaviors, such as pre-grasping under ungraspable conditions, tabletop rearrangement, and error recovery—bridging the gap between low-level skills and high-level autonomy.

Our main contributions are three-fold: (**i**) We propose the first RL-free framework for learning Bimanual Non-Prehensile Manipulation Primitives directly from human video demonstrations. (**ii**) We introduce a parameterization scheme that enables category-level generalization of non-prehensile skills across diverse object instances. (**iii**) We demonstrate the effectiveness, efficiency, versatility, and generality of BiNoMaP across a variety of tasks, objects, and strong baselines.

## 2 RELATED WORKS

**Non-Prehensile Manipulation** has long been recognized as a crucial topic in robotic learning Lynch & Mason (1999); Mason (1999); Zito et al. (2012), particularly for scenarios where grasping is infeasible. Prior to the rise of deep reinforcement learning (RL), traditional approaches predominantly relied on planning-based algorithms, such as graph search Maeda et al. (2001); Hou & Mason (2019); Cheng et al. (2022); Liang et al. (2023) or gradient-based optimization Posa et al. (2014); Moura et al. (2022); Xu et al. (2025b). However, these methods suffer from high computational cost and require precise physical priors (e.g., mass or friction coefficients), limiting their practical applicability. Recent advances have therefore shifted towards RL-based solutions, which bypass explicit planning by directly mapping sensory inputs to control actions, even in the presence of complex, contact-rich dynamics. Some works model manipulated objects using simplified geometric abstractions such as cylinders Lowrey et al. (2018), cuboids Yuan et al. (2018); Ferrandis et al. (2023), or bounding boxes Kim et al. (2023), enabling robust hybrid force-position control but at the cost of generalization to novel shapes. Others learn control policies that can generalize across shapes, yet they typically target single primitives like pushing Zhou et al. (2019); Zhong et al. (2025) or pivoting Zhang et al. (2023). More recent frameworks aim to learn a broad repertoire of non-prehensile skills under a unified RL architecture, as seen in CORN Cho et al. (2024), HACMan++ Jiang et al. (2024a), and HAMNET Cho et al. (2025). Additionally, the use of external dexterity—such as walls, edges, or ramps Yang et al. (2024); Wu et al. (2024); Wang et al. (2025)—is a common assumption to compensate for limited control authority, while tactile sensing has been explored to infer precise contact states Oller et al. (2024); Ferrandis et al. (2024); Shirai et al. (2025). To mitigate sim-to-real gaps, methods such as PIN-WM Li et al. (2025a) and DyWA Lyu et al. (2025) incorporate world models to reduce reliance on idealized observations and complete physical laws. Nonetheless, RL-based methods remain *fundamentally constrained by training instability and sample inefficiency*. Moreover, all existing studies adopt *a single-arm setup and heavily depend on non-generalizable environmental assumptions*. In contrast, our work challenges these limitations by introducing a more flexible and universal dual-arm setup, coupled with a three-stage RL-free learning paradigm.

**Bimanual Robotic Manipulation** often focuses on graspable skills, such as cloth-folding Colomé & Torras (2018), bagging Bahety et al. (2023), handover Li et al. (2023), untwisting Lin et al. (2024), dressing Zhu et al. (2024) and large behavior models Barreiros et al. (2025). For general bimanual manipulation, typical research Mirrazavi Salehian et al. (2016); Krebs & Asfour (2022); Zhao et al. (2023b) tends to explicitly classify them into uncoordinated and coordinated according to task characteristics. Most recently, the ALOHA series Zhao et al. (2023a); Fu et al. (2024); Zhao et al. (2024) have revolutionized bimanual manipulation by dexterous teleoperating and upgrading low-cost hardwares of real-world robots. These similar works Team et al. (2024); Kim et al. (2024); Liu et al. (2025b); Barreiros et al. (2025) implicitly train an end-to-end Vision-Language-Action (VLA) model using massive and diverse teleoperated data, expecting to get generalized robotic models. Instead of focusing on grasp-centric bimanual tasks, only a few studies address the dual-arm non-prehensile problem. For example, (Liu et al., 2022) targets a cooking scenario using the stir-fry skill. (Wu & Kruse, 2024) employs a hybrid dual-arm setup combining a gripper and a suction cup to handle shelf-based object picking. (Yamada et al., 2025) introduces a master-slave coordination scheme to accomplish constrained grasping. Other works address relatively constrained tasks such as bimanual ball lifting Grotz et al. (2024); Lu et al. (2025); Liu et al. (2025a). However, these approaches either focus on task-specific behaviors or rely on mixed hardware settings, and *do not aim to study bimanual non-prehensile skills methodically*. Moreover, the non-prehensile manipulation demands fine-grained interaction control that is difficult to achieve with ordinary dual-arm teleoperation systems—particularly those lacking force or tactile feedback Zhao et al. (2023a). As a result, collecting high-quality demonstrations for contact-intensive non-prehensile behaviors becomes impractical, which in turn hinders the scalability of data-driven imitation learning Chi et al. (2023); Black et al. (2025). In contrast, our work proposes a unified and scalable framework for learning generalizable bimanual non-prehensile skills from videos, without requiring laborious teleoperation, or deliberate hardware modifications.

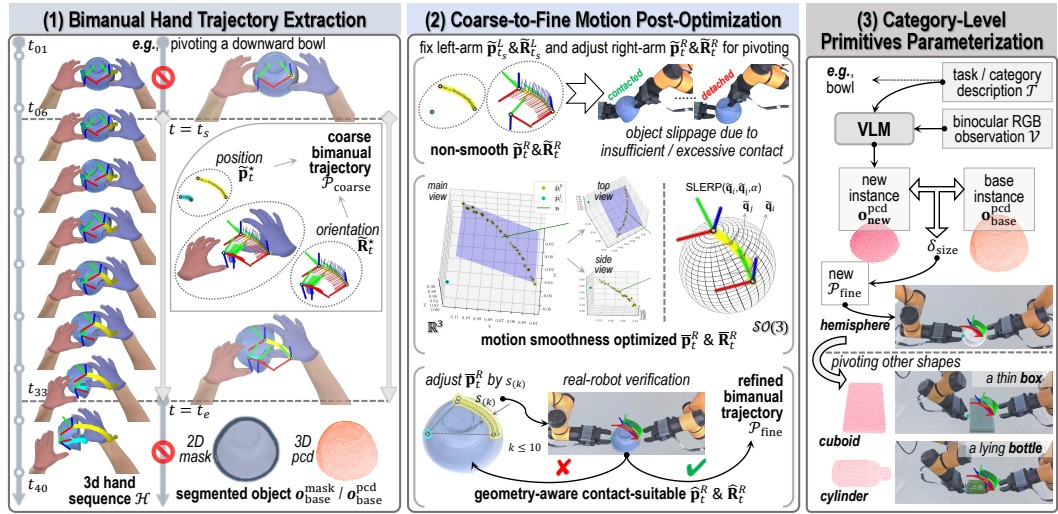

Figure 2: **BiNoMaP**. (1) The first stage leverages strong priors from hand demonstrations to obtain coarse dual-arm trajectories for non-prehensile tasks. (2) The second stage refines these trajectories to mitigate multi-source noise and improve execution stability. (3) The final stage generalizes learned skills to novel objects within the same category by parameterizing primitives.

## 3 METHOD

Our proposed RL-free learning framework BiNoMaP (Fig. 2) comprises three sequential stages: *Bimanual Hand Trajectory Extraction* (Sec. 3.2), *Coarse-to-Fine Motion Post-Optimization* (Sec. 3.3), and *Category-Level Primitive Parameterization* (Sec. 3.4). In the following, we first formalize our problem setup, and then elaborate on three stages in detail.

### 3.1 PROBLEM FORMULATION

**Bimanual Setting and Skill Categorization.** We target learning bimanual non-prehensile manipulation directly from human demonstrations without reinforcement learning, thereby bypassing limitations of RL such as slow convergence, training instability, and sim-to-real accuracy loss. Let the robot be equipped with two arms, denoted as $\mathcal{A} = \{\mathcal{A}^L, \mathcal{A}^R\}$. We do not assume access to environmental constraints beyond manipulated objects themselves. Under this setup, we consider three major categories of non-prehensile skills: *(1) single-arm skills with handedness* — tasks executable with one arm but requiring left/right preference, e.g., pushing, poking; *(2) one arm manipulates, one stabilizes* — asymmetric cooperation, e.g., pivoting, occluded grasping. *(3) dual-arm cooperative motion* — simultaneous symmetric/asymmetric motion of both arms, e.g., wrapping, carrying.

**RL-free Learning Objective.** Learning robotic manipulation from human videos is promising yet challenging Li et al. (2024); Ko et al. (2024); Luo et al. (2025), and remains largely unexplored for non-prehensile skills. Our motivation is that human bimanual hand motion inherently encodes coordination patterns required for non-prehensile manipulation. By leveraging these strong priors and focusing on error correction rather than unconstrained exploration, we can construct an efficient RL-free imitation learning paradigm. Formally, let $\mathcal{T}$ means task description (text), $\mathcal{V}$ is RGB-D video of human demonstration, $\mathcal{O}$ represents visual observation of the manipulated object(s) (e.g., RGB or 3D point cloud), and $\mathcal{A}$ indicates a dual-arm robot. We aim to learn a mapping:

$$f_\theta : (\mathcal{T}, \mathcal{V}, \mathcal{O}, \mathcal{A}) \longrightarrow \mathcal{P}, \tag{1}$$

where $\mathcal{P} = \{\mathbf{p}_t, \mathbf{R}_t\}_{t=1}^T$ is the time-parameterized primitive trajectory, with $\mathbf{p}_t \in \mathbb{R}^3$ denoting the end-effector position and $\mathbf{R}_t \in SO(3)$ the end-effector orientation at time $t$. The gripper states are ignored for non-prehensile tasks. The following subsections detail how $\mathcal{P}$ is obtained.

### 3.2 BIMANUAL HAND TRAJECTORY EXTRACTION

**3D Hand Reconstruction from Videos.** We conduct human demonstrations on a table within the dual-arm workspace. For each target skill, a human demonstrator executes the motion once while a

stereo camera records RGB streams, with the left view $\mathcal{V}^L$ used as the main observational source. We ensure both hands remain visible. Since only contact-relevant motion segments are necessary for primitive extraction, we manually specify start and end frames $(t_s, t_e)$ — e.g., from hands first touch the object to hands release the object — removing redundant approach/retreat phases, which can be addressed by motion planning during real robots deployment. For each frame $t \in [t_s, t_e]$ among $T'$ frames, we apply the 3D hand reconstruction algorithm WiLoR Potamias et al. (2025) to estimate both 3D hand shapes $\mathbf{M}_t^\star$ (the MANO Romero et al. (2017) parametric model capturing pose and shape parameters) and the corresponding handedness $\star \in \{L, R\}$ (indicating left/right hand). This finally yields a hand attribute sequence $\mathcal{H} = \{\mathbf{M}_t^\star\}_{t=t_s}^{T'}$.

**Hand-to-Robot Trajectory Extraction.** To retarget hands to parallel-jaw grippers, we simplify each hand's mesh in $\mathcal{H}$ to a representative contact point: the midpoint between the thumb tip and index finger tip among 21 predefined joints by MANO. Because WiLoR outputs per-frame meshes in independent coordinates (no camera intrinsics), we thus first project the contact points to 2D image space, associate them with the 3D scene point cloud, and then transform into a unified camera coordinate frame to obtain $\tilde{\mathbf{p}}_t^\star \in \mathbb{R}^3$. To estimate the end-effector orientation, we approximate each hand's 3D orienta-

---

**Algorithm 1** Approximation of 3D Hand Pose.

- **Input:** 3D hand shape $\mathbf{M}_t^\star$, 21 pre-defined 3D hand joints $I_{\text{hand}}$, indexs of wrist joint $i_{\text{wri}}$ / index-fingertip $i_{\text{ind}}$ / ring-fingertip $i_{\text{ring}}$, the given handedness $\star = L$ or $\star = R$.
- **Output:** 3D hand pose $\tilde{\mathbf{R}}_t^\star$.                    // either $L$ or $R$
Initialize $\mathbf{P}_t^\star \leftarrow \mathbf{MANO}(\mathbf{M}_t^\star, I_{\text{hand}})$;    // 3D hand joints array
$p_{\text{wri}} \leftarrow \mathbf{P}_t^\star[i_{\text{wri}}]$, $p_{\text{ind}} \leftarrow \mathbf{P}_t^\star[i_{\text{ind}}]$, $p_{\text{ring}} \leftarrow \mathbf{P}_t^\star[i_{\text{ring}}]$;
$l_{\text{iw}} \leftarrow (p_{\text{ind}} - p_{\text{wri}})$, $l_{\text{rw}} \leftarrow (p_{\text{ring}} - p_{\text{wri}})$;    // two 3D lines
$v_z \leftarrow \texttt{cross\_product}(l_{\text{iw}}, l_{\text{rw}})$;    // Z-direction
$\bar{v}_z \leftarrow v_z / (\texttt{normalize}(v_z) + 1\text{e-}8)$;    // normalize vector
$v_y = l_{\text{mid}} \leftarrow (l_{\text{iw}} + l_{\text{rw}}) / 2.0$;    // estimated Y-direction
$\bar{v}_y \leftarrow v_y / (\texttt{normalize}(v_y) + 1\text{e-}8)$;    // normalize vector
$\bar{v}_x \leftarrow \texttt{cross\_product}(\bar{v}_y, \bar{v}_z)$;    // X-direction
$\tilde{\mathbf{R}}_t^\star \leftarrow \texttt{concatenate}([\bar{v}_x, \bar{v}_y, \bar{v}_z])$;    // 3×3 rotation matrix
**return** $\tilde{\mathbf{R}}_t^\star$;

---

tion by computing a rotation matrix $\tilde{\mathbf{R}}_t^\star \in SO(3)$ as described in Alg. 1, which is analogous to the eigengrasping Ciocarlie et al. (2007) by aligning the index-ring fingertip direction with the gripper's spindle. The coarse bimanual trajectory is thus $\mathcal{P}_{\text{coarse}} = \{(\tilde{\mathbf{p}}_t^\star, \tilde{\mathbf{R}}_t^\star)\}_{t=t_s}^{T'}$.

For later refinement, we further extract the manipulated object's geometry from the first frame $t_s$. We first utilize vision-language models (e.g., Florence-2 Xiao et al. (2024) + SAM2 Ravi et al. (2025)) to obtain a 2D object mask $\mathbf{o}_{\text{base}}^{\text{mask}}$. Then, we map the mask to the scene's 3D point cloud and store the segmented object point cloud $\mathbf{o}_{\text{base}}^{\text{pcd}}$ as an input for the post-optimization stage. At this point, $\mathcal{P}_{\text{coarse}}$ is ready for refinement to address multi-source noise from reconstruction errors, stereo matching, and hand–robot morphology gaps.

### 3.3 Coarse-to-Fine Motion Post-Optimization

We perform post-optimization of initially extracted $\mathcal{P}_{\text{coarse}}$ from two complementary perspectives: *motion smoothness* and *appropriate object contact*. The former reduces spatial jitter by smoothing positional and rotational transitions. The latter ensures that end-effectors maintain stable and safe contact, preventing both object slippage due to insufficient contact and deformation or rebound due to excessive force. We note an important prerequisite for all skills: the trajectory points remain *coplanar* in each arm, regardless of whether the arms move synchronously or asynchronously.

**Motion Smoothness Optimization.** We first enforce positional smoothing with coplanarity constraint. For a single-arm trajectory $\{\tilde{\mathbf{p}}_t\}_{t=t_s}^{T'}$, we fit an optimal plane $\prod$ to them via least squares:

$$\arg \min_{\mathbf{n}, b} \sum_{t=t_s}^{T'} (\mathbf{n}^\top \tilde{\mathbf{p}}_t + b)^2, \quad \texttt{s.t.} \ \|\mathbf{n}\|_2 = 1, \tag{2}$$

where $\mathbf{n}$ is the plane normal and $b$ is the offset. All trajectory points are then orthogonally projected onto $\prod$, and 2D trajectory smoothing filter (e.g., cubic B-spline De Boor (1978)) is applied in the plane coordinates to penalize curvature and enforce local smoothness.

Similarly, we smooth the sequence of rotation matrices $\{\tilde{\mathbf{R}}_t\}_{t=t_s}^{T'}$ in $SO(3)$ for skills involving significant orientation changes. We first select a set of anchor frames $\mathcal{K} \subset \{t_s, \cdots, t_e\}$—including the start, the end, and top-$n$ intermediate frames with minimal positional error—to serve as orientation

constraints. Between anchors, we apply spherical linear interpolation (SLERP) Shoemake (1985) to ensure smooth rotational transitions aligned with the refined positions:

$$\tilde{\mathbf{q}}(\alpha) = \mathsf{SLERP}(\tilde{\mathbf{q}}_i, \tilde{\mathbf{q}}_j; \alpha) = \tilde{\mathbf{q}}_i(\tilde{\mathbf{q}}_i^\top \tilde{\mathbf{q}}_j)^\alpha, \quad i, j \in \mathcal{K}, \alpha \in [0, 1], \tag{3}$$

where $\tilde{\mathbf{q}}_i$ and $\tilde{\mathbf{q}}_j$ are quaternions of matrices $\tilde{\mathbf{R}}_i$ and $\tilde{\mathbf{R}}_j$ from two selected adjacent anchors. The uniform sampling frequency of $\alpha$ depends on the number of original intermediate points omitted. The computed quaternion $\tilde{\mathbf{q}}(\alpha)$ will be converted back into a rotation matrix $\tilde{\mathbf{R}}(\alpha)$. All optimizations are performed offline, making them computationally efficient.

**Geometry-Aware Iterative Contact Adjustment.** After obtaining coarse yet smooth trajectories $\{\overline{\mathbf{p}}_t^L\}_{t=t_s}^{T'}$ and $\{\overline{\mathbf{p}}_t^R\}_{t=t_s}^{T'}$ for both arms, we refine contact geometry to ensure appropriate interaction with the object. This is achieved through an iterative, geometry-aware real-robot verification process. Without loss of generality, let the right/left arm be the primary/support arm. In iteration $k$, we first adjust the initial contact point $\overline{\mathbf{p}}_{t_s}^R$ towards the object $\mathbf{o}_{\mathrm{base}}^{\mathrm{pcd}}$ to achieve a target distance $d_{(k)}$. This yields an adjusted point $\overline{\mathbf{p}}_{t_s,(k)}^R$. This adjustment defines a scaling factor for the relative motion between two arms: $s_{(k)} = \|\overline{\mathbf{p}}_{t_s,(k)}^R - \overline{\mathbf{p}}_{t_s}^L\|_2 / \|\overline{\mathbf{p}}_{t_s}^R - \overline{\mathbf{p}}_{t_s}^L\|_2$. The scaling factor is then used to update the entire primary arm trajectory, while the support arm trajectory remains unchanged:

$$\overline{\mathbf{p}}_{t,(k)}^R = \overline{\mathbf{p}}_t^L + s_{(k)}(\overline{\mathbf{p}}_t^R - \overline{\mathbf{p}}_t^L), \quad \forall t \in [t_s, t_e]. \tag{4}$$

We initialize the verification with a large distance $d_{(1)} = 5\mathrm{mm}$ for keeping safety. If the manipulation fails, we iteratively decrease this distance $d_{(k)} = d_{(1)} \cdot \gamma^{(k-1)}$ (e.g., $\gamma = 0.85$) and re-evaluate, up to a maximum of ten attempts ($k \leq 10$). The rationality of related hyper-parameters will be verified in ablation studies. This process, which robustly applies to diverse skills like pivoting and dual-arm wrapping, typically converges to a successful trajectory taking less than five minutes (Note that this is an one-time real-world coarse-to-fine refinement for each new object). We denote the final refined bimanual trajectory as $\mathcal{P}_{\mathrm{fine}} = \{(\hat{\mathbf{p}}_t^\star, \hat{\mathbf{R}}_t^\star)\}_{t=t_s}^{T'}$, which can be treated as an instance-level primitive skill. This approach is also significantly more efficient and stable than simulation-based RL Cho et al. (2024); Lyu et al. (2025), and safer than real-world RL Luo et al. (2024a;b).

Furthermore, during deployment, object relocation is handled by detecting the target with VLMs Xiao et al. (2024); Ravi et al. (2025), computing its planar displacement $(\Delta x, \Delta y)$, and applying a corresponding translation to the refined atomic trajectory, enabling spatial generalization.

### 3.4 CATEGORY-LEVEL PRIMITIVE PARAMETERIZATION

To move beyond manipulating a single object instance—a common limitation of RL-based methods—we parameterize each optimized atomic trajectory to create a category-level primitive. Our geometry-aware optimization from Sec. 3.3 implicitly encodes object dimensions. We make this explicit to allow the skill to adapt to other objects of the same category.

The core strategy is to treat the object in the initial optimization as the base instance $\mathbf{o}_{\mathrm{base}}^{\mathrm{pcd}}$. For any new instance, we compute its dimensional variation relative to this base. This is done by acquiring the new object point cloud $\mathbf{o}_{\mathrm{new}}^{\mathrm{pcd}}$, taking a horizontal slice at the initial contact height, and calculating a characteristic dimension (e.g., the maximum point cloud distance for a bowl diameter in pivoting):

$$\delta_{\mathrm{size}} = \max_{\mathbf{u},\mathbf{v}\in\mathbf{o}_{\mathrm{new}}^{\mathrm{pcd}}} \|\mathbf{u} - \mathbf{v}\|_2 - \max_{\mathbf{u},\mathbf{v}\in\mathbf{o}_{\mathrm{base}}^{\mathrm{pcd}}} \|\mathbf{u} - \mathbf{v}\|_2, \quad \texttt{s.t.} \ (\mathbf{u} - \mathbf{v}) \parallel (\hat{\mathbf{p}}_{t_s}^L - \hat{\mathbf{p}}_{t_s}^R). \tag{5}$$

The size difference $\delta_{\mathrm{size}}$ between the new and base instances is then incorporated into our contact optimization Eqn. 4 in a single, non-iterative step to adapt the trajectory. The corresponding verified target distance $d_{(\hat{k})}$ and scaling factor $s_{(\hat{k})}$ are modulated by $\delta_{\mathrm{size}}$ to adjust the inter-arm distance, effectively resizing the manipulation primitive $\mathcal{P}_{\mathrm{fine}}$ for the new object. This parameterization allows BiNoMaP to apply a learned skill to a wide range of intra-category objects without requiring new human demonstrations or repeating the full extraction and optimization pipeline. This significant advantage in scalability and convenience will be demonstrated in our experiments.

## 4 EXPERIMENTS

We aim to address four central questions in our experiments. (**Q1**): Does BiNoMaP demonstrate significant advantages over state-of-the-art visuomotor-based or RL-based baselines? (**Q2**): Are all

Figure 3: Examples of four non-prehensile skills instantiated with different tasks and diverse objects.

Table 1: Quantitative comparison results of our BiNoMaP and six baselines under three skills. All training times were collected on a single RTX 4090 with 24GB memory.

| Methods | Year | Policy Type | Training Time (h) | *poking* (L/R) | | *pivoting* (LR) | | *wrapping* (LR) | | Average Success Rate |
|---|---|---|---|---|---|---|---|---|---|---|
| | | | | plastic cup | ceramic mug | plastic bowl | papery box | smooth ball | inverted basket | |
| ACT Zhao et al. (2023a) | RSS'23 | Visuomotor | 8.5 | 10/30 | 03/30 | 01/30 | 05/30 | 11/30 | 01/30 | 17.2% (31/180) |
| DP Chi et al. (2023) | RSS'23 | Visuomotor | 4.8 | 16/30 | 11/30 | 00/30 | 10/30 | 20/30 | 04/30 | 33.9% (61/180) |
| DP3 Ze et al. (2024) | RSS'24 | Visuomotor | 6.2 | 20/30 | 14/30 | 04/30 | 14/30 | 23/30 | 06/30 | 45.0% (81/180) |
| HACMan Zhou et al. (2023) | CoRL'23 | RL-based | 11.1 | 07/30 | 00/30 | 01/30 | 02/30 | 07/30 | 02/30 | 10.6% (19/180) |
| CORN Cho et al. (2024) | ICLR'24 | RL-based | 24.0 | 16/30 | 15/30 | 01/30 | 11/30 | 17/30 | 06/30 | 36.7% (66/180) |
| DyWA Lyu et al. (2025) | ICCV'25 | RL-based | 41.2 | 21/30 | 19/30 | 04/30 | 17/30 | 20/30 | 07/30 | 48.9% (88/180) |
| BiNoMaP (ours) | — | RL-free | 0.1 | 29/30 | 25/30 | 20/30 | 26/30 | 30/30 | 25/30 | **86.1% (155/180)** |

the components in the multi-phase modular design of BiNoMaP necessary and beneficial? (Q3): Can the proposed BiNoMaP framework generalize across a wide range of non-prehensile skills and rapidly adapt to diverse object shapes? (Q4): Can the skills learned by BiNoMaP serve as effective building blocks for accomplishing higher-level downstream manipulation tasks?

## 4.1 EXPERIMENT SETUPS AND PROTOCOL

**Tasks and Setups.** We consider four typical non-prehensile skills (see Fig. 1), each of which can be instantiated on different objects to complete diverse tasks. These include *uprightting a toppled cup*, *flipping over an upside-down bowl*, *lifting a thin cuboid box to a standing pose*, *translating a smooth ball*, *reorienting a face-down basket to face-up*, etc. (see Fig. 3). We adopt a **dual-arm setup** that eliminates the reliance on external affordances. This design leverages the intrinsic advantages of bimanual manipulation, including left–right hand complementarity, fixed-moving collaboration, and synchronous coordination. Detailed specifications of the manipulation platform, hardware components, object assets, and exact tasks covered by each skill are provided in Appendix A.

**Baselines and Metric.** To objectively evaluate the superiority of BiNoMaP, we conduct quantitative comparisons on three skills—*poking*, *pivoting*, and *wrapping*—against six strong baselines (including ACT Zhao et al. (2023a), DP Chi et al. (2023), 3DP Ze et al. (2024), HACMan Zhou et al. (2023), CORN Cho et al. (2024), and DyWA Lyu et al. (2025)). The first two are visuomotor policies learned from real robot demonstrations, while the latter two are RL-based sim-to-real methods. We mainly compare the **Success Rates** of executing each skill on the same object placed at different positions. Unless otherwise specified, each object–task pair is evaluated with 30 real-world trials. Details of all reproduced baselines are included in Appendix B. In addition, to thoroughly evaluate the performance of BiNoMaP, we further perform ablation studies on its modular components, as well as category-level generalization tests of each learned skill.

## 4.2 RESULTS COMPARISON AND ANALYSIS

**(A1) Comparison with Visuomotor and RL-based Baselines.** As shown in Tab. 1, our BiNoMaP significantly outperforms advanced non-prehensile manipulation methods across six tasks spanning three representative skills, when compared against two classes of baselines—visuomotor imitation policies and RL-based policies. Although these baselines are trained and tested on the same fixed object instances, their performance remains suboptimal. In particular, the two most challenging tasks, *pivoting a plastic bowl* and *wrapping an inverted basket*, require precise bimanual coordination and

Table 2: Ablation studies of our method BiNoMaP.

| Critical Components | | | | Success Rate | |
|---|---|---|---|---|---|
| points mapping | points smoothing | pose smoothing | contact adjusting | *poking* bowl | *wrapping* basket |
| ✗ | ✓ | ✓ | ✓ | 07/20 | 08/20 |
| ✓ | ✗ | ✓ | ✓ | 10/20 | 12/20 |
| ✓ | ✓ | ✗ | ✓ | 11/20 | 12/20 |
| ✓ | ✓ | ✓ | ✗ | 13/20 | 14/20 |
| ✓ | ✓ | ✓ | ✓ | **15/20** | **17/20** |

Table 3: Ablation on hyper-parameters (taking *poking bowl* as an example).

| top-$n$ | 1 | 2 | **3** | 4 | 5 |
|---|---|---|---|---|---|
| SR | 05/10 | 06/10 | **07/10** | **07/10** | 06/10 |
| $d_{(1)}$(mm) | 3 | 4 | **5** | 6 | 7 |
| SR | 06/10 | 06/10 | **07/10** | 05/10 | 05/10 |
| factor $\gamma$ | 0.75 | 0.80 | **0.85** | 0.90 | 0.95 |
| SR | 06/10 | **07/10** | 07/10 | 06/10 | 05/10 |

Table 4: Quantitative results of our BiNoMaP and baselines on all four non-prehensile skills. Trails are adjusted into 10 for each category-level instance under those unseen objects testing.

| Methods | Generalization | *poking* (L/R) | | *pivoting* (LR) | | | *pushing* (LR) | *wrapping* (LR) | | Average Success Rate |
|---|---|---|---|---|---|---|---|---|---|---|
| | **Seen Objects** | plastic cup | ceramic mug | plastic bowl | papery box | plastic bottle | heavy basket | smooth ball | inverted basket | |
| Ours | Instance-Level | 29/30 | 25/30 | 20/30 | 26/30 | 25/30 | 28/30 | 30/30 | 25/30 | **86.7%** |
| | **Unseen Objects** | other 6 cups | other 6 mugs | other 7 bowls | other 7 boxes | other 7 bottles | other 5 baskets | other 2 balls | other 5 baskets | |
| DP3 | Category-Level | 26/60 | 17/60 | 03/70 | 22/70 | — | — | 11/20 | 05/50 | 25.5%(19.5% ↓) |
| DyWA | Category-Level | 30/60 | 20/60 | 06/70 | 26/70 | — | — | 10/20 | 06/50 | 29.7%(18.6% ↓) |
| Ours | Category-Level | 51/60 | 43/60 | 46/70 | 55/70 | 51/70 | 43/50 | 17/20 | 37/50 | **76.2%**(10.1% ↓) |

continuous execution of contact-rich actions. All six baselines frequently suffer from either insufficient or excessive contact, leading to premature object slippage and consequently very low success rates. For the remaining tasks, higher success rates can be partially attributed to favorable dynamics, such as leveraging friction between end-effectors and objects, or exploiting gravitational torque balance (e.g., *hooking the inner wall of a mug to lift it upward*, or *supporting a sphere from below with both arms*). However, these approaches either rely on direct perception-to-action mappings without explicit contact reasoning (visuomotor policies ACT, DP and DP3) or incur performance degradation during sim-to-real transfer (RL-based HACMan, CORN and DyWA), and thus fail to explicitly address the requirement of frequent and fine-grained contact handling. In contrast, BiNoMaP leverages VLMs to localize novel object placements and segment 3D point clouds, followed by adaptive adjustment of learned instance-level skills. This design yields both interpretability and robustness, achieving an average success rate of **86.1%** across all tasks—approximately twice that of latest strong baselines—demonstrating its high practicality and effectiveness.

**(A2) Ablation Studies on Modular Design.** To assess the effectiveness of BiNoMaP's modular design, we conduct ablation studies on two tasks *pivoting bowl* and *wrapping basket* (see Tab. 2). Specifically, we evaluate the contribution of three critical components: (i) the trajectory extraction stage that approximates bimanual motion via mapping 3D hand representative points $\mapsto$ 2D pixel indices $\mapsto$ 3D object point cloud; (ii) the post-optimization stage with 3D point smoothing and 3D pose interpolation; and (iii) the iterative contact adjustment strategy. Trails are adjusted into 20. Results show that if we skip the projection-and-indexing approximation and instead rely solely on estimated 3D hand points per frame, the success rate drops drastically, even after full post-optimization—highlighting the necessity of our trajectory extraction strategy. Moreover, removing either the 3D point smoothing or 3D pose interpolation module leads to a notable performance decrease, confirming their role in eliminating jitter in both position and orientation dimensions, and thus maintaining stable contact between the object and end-effectors. Similarly, discarding the iterative contact adjustment also lowers success rates, indicating its indispensability for millimeter-level precision in contact-rich skills such as *pivoting* and *wrapping*, and by intuition, its utility for other non-prehensile skills as well. Furthermore, as shown in Tab. 3, hyper-parameter sweeps over the number of intermediate frames ($n$), the initial contact distance ($d_{(1)}$), and the decay factor ($\gamma$) reveal only minor fluctuations in success rates, suggesting that BiNoMaP is robust and insensitive to hyper-parameter choices, ensuring broad adaptability across different skills and objects.

**(A3) Generalization Across Skills and Objects.** To evaluate the generalizability of BiNoMaP, we conduct real-world experiments across four non-prehensile skills and a total of eight object-task pairs (see Tab. 4 and Fig. 3). Results show that BiNoMaP applies broadly to diverse non-prehensile manipulation skills: under *instance-level generalization* (same object, varying placements), the framework achieves an average success rate of nearly **86.7%**, covering objects with cylindrical, hemispherical, cuboidal, and spherical geometries, as well as materials with different frictional properties such

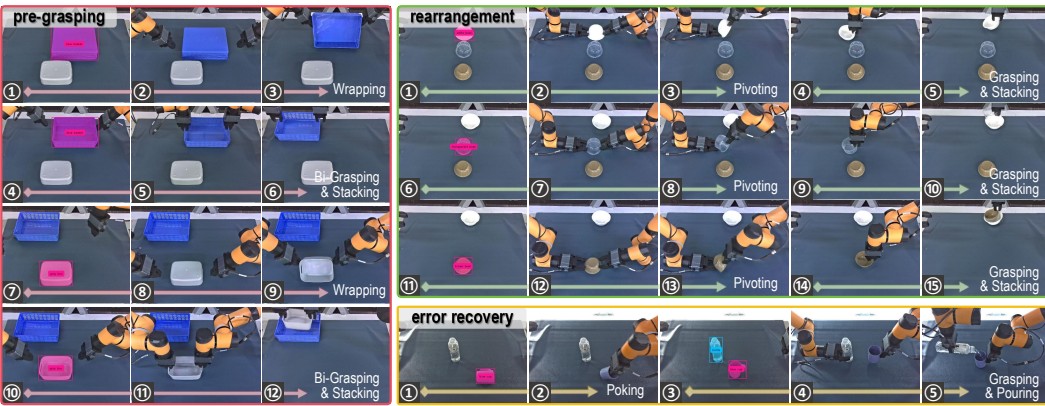

Figure 4: Examples of utilizing learned non-prehensile skills to boost complex manipulation tasks.

as plastic, ceramic, and paperboard. More importantly, under the more challenging *category-level generalization* setting (varying placements and object sizes), BiNoMaP still maintains a promising success rate of around **76.2%**, with only a modest performance drop. In contrast, existing visuomotor and RL-based baselines perform poorly even in instance-level generalization (30–50% success; see Tab. 1), and inevitably deteriorate further under out-of-distribution category-level tests. While such limitations could be partially alleviated by collecting more demonstrations or prolonging training in simulation, these approaches are costly, difficult to scale, and not necessarily effective for contact-rich non-prehensile tasks. By comparison, BiNoMaP offers a more *scalable and deployment-friendly* solution for real-world applications.

**(A4) Compositionality for Downstream Tasks.** A key advantage of BiNoMaP lies in the modularity and transferability of its learned atomic skills, which can be seamlessly integrated into higher-level downstream manipulation tasks. We explore three representative applications that highlight this property: (1) using the *wrapping basket* skill to flip an inverted basket upright for subsequent bimanual grasping (**pre-grasping**); (2) applying the *pivoting bowl* skill to sequentially flip and stack multiple inverted non-prehensile bowls on the table (**rearrangement**); and (3) employing the *poking mug* skill to upright a fallen mug so that filling water poured from a bottle can be performed afterward (**error recovery**). In each case, once the object is manipulated into a graspable state by BiNoMaP, we leverage VLMs to localize and segment the object, apply AnyGrasp Fang et al. (2023) to obtain 6-DoF grasp poses, and execute motion planning via inverse kinematics. As illustrated in Fig. 4, all three cases were validated in real-world experiments under the same hardware setup, yielding consistently successful results. These demonstrations underscore the composability and practical utility of BiNoMaP, and point toward promising future directions where the framework can be embedded into longer-horizon and more complex manipulation pipelines.

## 5 CONCLUSION

In this work, we presented BiNoMaP, a novel framework for learning category-level bimanual non-prehensile skills. Through extensive real-world experiments, we demonstrated that BiNoMaP not only significantly outperforms strong visuomotor and RL-based baselines, but also generalizes effectively across diverse skills and object shapes. Its modular design was shown to be both effective and robust, with each component contributing positively to performance. Moreover, the learned primitives were successfully composed into higher-level downstream tasks, revealing strong composability and practical applicability. Taken together, these results highlight BiNoMaP as a promising step toward scalable and generalizable non-prehensile bimanual manipulation.

**Limitations**. First, the current framework operates in an open-loop manner and lacks closed-loop correction, making it vulnerable to execution errors or contact deviations without re-observation and re-planning. Second, without tactile sensing, BiNoMaP is less effective for manipulating strictly rigid objects with minimal compliance (e.g., pivoting metal, ceramic or glass bowls/plates), where fine-grained force feedback is critical. Finally, BiNoMaP is now designed in a skill- or category-specific manner, whereas integrating it into a more unified language-driven VLA model that schedules multiple skills end-to-end would be an attractive solution for real-world applications.

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

APPENDIX

This appendix provides additional details to complement the main text. Sec. A (**Specifications of Tasks and Setups**) formalizes the definitions and experimental setups of non-prehensile tasks, offering a clearer understanding of the problem space. Sec. B (**Reproduction of All Six Baselines**) reviews the six advanced baselines and explains our reproduction settings to ensure fair comparison. Sec. C (**More Details and Results of BiNoMaP**) presents extended methodological and experimental details of BiNoMaP, facilitating deeper technical insights and reproducibility. Sec. D (**Summary and Analysis of Failure Cases**) conducts a statistical analysis of failure cases across different skills and tasks, highlighting the challenges and limitations. Sec. E (**Cross-Embodiment Transferability of BiNoMaP**) reveals the cross-platform deployment of learned primitive skills, indicating their fantastic reusability. Sec. F (**Details of Integration with Downstream Applications**) demonstrates how BiNoMaP can be seamlessly integrated into higher-level downstream tasks, further showcasing its versatility and application potential. Sec. G (**Further Verification of Generalization and Adaptability**) includes more generalization experiments to further validate the adaptability of BiNoMaP. Finally, Sec. H is the statement on the use of LLMs.

## A  SPECIFICATIONS OF TASKS AND SETUPS

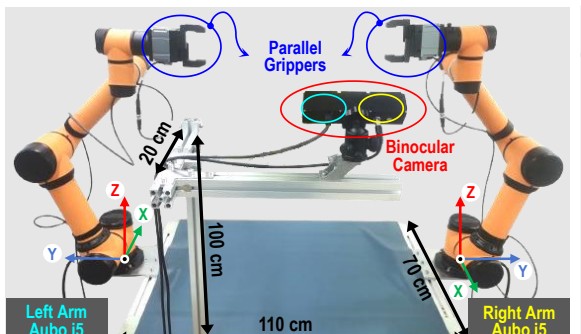

| Skill: Object (Task) | Gripper Open Percentage | Approximate Object Shapes |
|---|---|---|
| Poking: Cup | 0% | Cylinder |
| Poking: Mug | 0% | Cylinder |
| Pivoting: Bowl | 50% | Hemisphere |
| Pivoting: Box | 50% | Cuboid |
| Pivoting: Bottle | 25% | Cylinder |
| Pushing: Basket | 0% | Cuboid |
| Wrapping: Ball | 50% | Spherical |
| Wrapping: Basket | 0% | Cuboid |

Figure 5: *Left*: The fixed-base dual-arm manipulator platform used in this research. *Right*: The grippers opening ratios for each skill and task.

### A.1  HARDWARE AND DUAL-ARM PLATFORM

Our experimental platform is a rectangular table measuring approximately 110 cm in length and 70 cm in width, equipped with a dual-arm robotic system, parallel grippers, and a binocular observation camera (see Fig. 5 left for layout). The two arms are mounted on opposite sides of the short edges of the table—different from the more common same-side or humanoid-like configurations. This design greatly reduces workspace overlap and improves reachability, though it departs from human arm kinematics and behaviors. Each arm is a 6-DOF Aubo-i5 collaborative manipulator[1] (without joint-level force control) with a maximum reach of about 880 mm. The end-effectors are DH-Robotics parallel grippers[2] with an 80 mm maximum opening, an effective length of 50 mm, and a total length of 160 mm (compensated in flange length modeling).

Since we focus on bimanual non-prehensile manipulation, the two grippers remain unchanged during execution and do not open or close; their opening ratios vary across skills and object shapes, as summarized in Fig. 5 right. To increase friction at the metallic tips, black electrical tape was applied around all four tips, which did not negatively affect grasp-based tasks as revealed in our downstream manipulation experiments. For perception, we use a Kingfisher R-6000 binocular camera, with RGB images resized to 960×540, supporting calibrated stereo matching and reconstruction of 3D point clouds. This setup provides quality comparable to or exceeding standard RGB-D sensors, with the added flexibility of customized algorithms. The camera is mounted in a third-person view, fixed on one long edge of the table at 20 cm offset and 100 cm height, covering nearly the entire tabletop. This configuration suffices for both recording human demonstration videos and executing all dual-arm non-prehensile tasks, making eye-in-hand cameras unnecessary.

---

[1]https://www.aubo-cobot.com/public/i5product3
[2]https://en.dh-robotics.com/product/pg

## A.2 SKILL AND TASK FORMULATION

To comprehensively evaluate BiNoMaP's capability in dual-arm non-prehensile manipulation, we select four representative skills—*poking*, *pivoting*, *pushing*, and *wrapping*—which collectively cover left/right hand specialization, inter-arm coordination, and synchronous motion requirements. Based on these skills, we further design eight concrete dual-arm non-prehensile tasks, each involving a category of objects with diverse shapes and sizes (see Fig. 6). This diversity facilitates assessing the generalization ability of policies across different object instances. Below we describe the objectives of each skill and its associated tasks in detail:

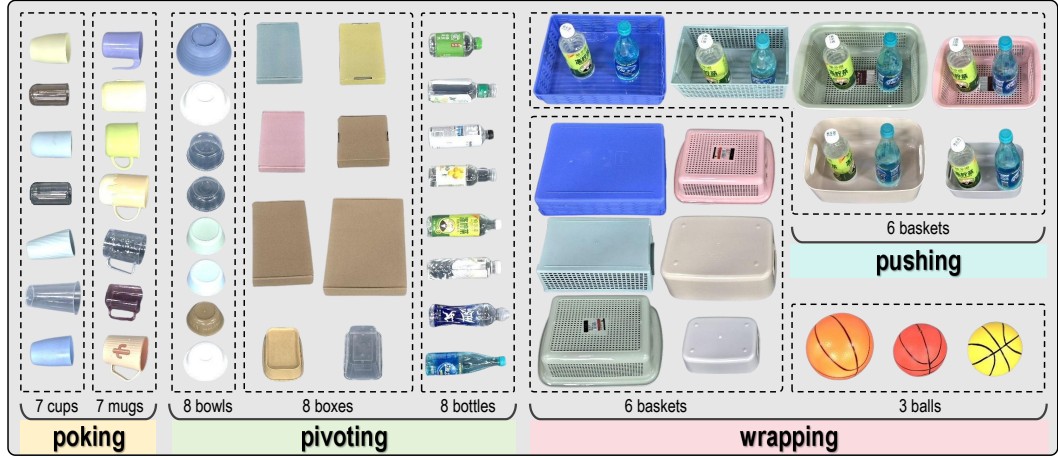

Figure 6: The object assets involved in our selected four non-prehensile skills and eight bimanual manipulation tasks. All objects have been scaled down proportionally.

- ***poking***. *This skill uses one arm's end-effector to lift a horizontally lying object back to an upright pose, requiring explicit left–right handedness.*
  Two tasks are defined: **poking cup** (7 cups without handles, with nearly uniform mass distribution) and **poking mug** (7 mugs with handles of varying shapes, positions, and sizes, leading to non-uniform and shifting mass centers during manipulation). For consistency, objects are placed such that their openings face either the left or right arm base; the evaluation considers only positional generalization. Unlike prior non-prehensile studies (e.g., CORN Cho et al. (2024), DyWA Lyu et al. (2025), PIN-WM Li et al. (2025a)) that "poke" the bottom of cups, we deliberately avoid this approach, as it risks damaging objects and requires excessive taping of the tooltips—compromising subsequent grasping tasks. Instead, we adopt an inside-lip lifting motion, which is safer and more generalizable.

- ***pivoting***. *This skill stabilizes an object with one arm (serving as a "wall" role, akin to extrinsic dexterity assumptions in previous works) while the other arm presses from the opposite side and rotates the object upward around the stabilizing point, typically by ~90°. This requires fine dual-arm coordination.*
  Three tasks are included: **pivoting bowl** (8 hemispherical bowls placed upside-down, goal: flip them upright), **pivoting box** (8 thin rectangular boxes, initially flat and empty, goal: stand them vertically on the shortest edge), and **pivoting bottle** (8 cylindrical bottles with varying water contents to avoid instability, goal: place them upright with the cap upward). During experiments, bowls require no orientation constraint; boxes are placed with their long side perpendicular to the arm baseline; bottles always have caps facing the right arm. These initial states can be reasonably prepared by 6D pose estimation plus planar pushing, hence not the focus here. To our knowledge, while pivoting boxes have been previously studied Sun et al. (2020); Zhou & Held (2023); Zhang et al. (2023); Yamada et al. (2025); Wang et al. (2025), pivoting bowls and bottles are novel in non-prehensile manipulation research.

- ***pushing***. *This skill involves both arms simultaneously approaching a large-size object from one side and pushing it along a fixed direction over a given distance, emphasizing synchronized dual-arm motion.*
  One single task is defined: **pushing basket** (6 baskets with heavy contents inside). The objective is to push the basket slowly and smoothly by ~20 cm without causing significant displacement of the items inside. To sensitively reflect performance differences of various manipulation

policies, two water bottles are placed in each basket; if either bottle falls during manipulation, the task is considered failed.

- ***wrapping***. *This skill requires the two arms to approach an object from opposite sides simultaneously, enclose it, and perform a lift–translate–place motion sequence, testing both coordination and synchronization.*

  Two tasks are included: **wrapping ball** (3 balls of different sizes, goal: securely enclose and transfer the ball without dropping, essentially a bimanual pick-and-place) and **wrapping basket** (6 upside-down baskets without protrusions or handles, goal: lift from one side, let gravity expose the opening, then use the arms' enclosing workspace to reorient and place the basket upright, achieving a 180° flip). The wrapping-ball task has appeared in prior works such as AnyBimanual Lu et al. (2025) and D-CODA Liu et al. (2025a), while wrapping-basket is, to our knowledge, introduced here for the first time.

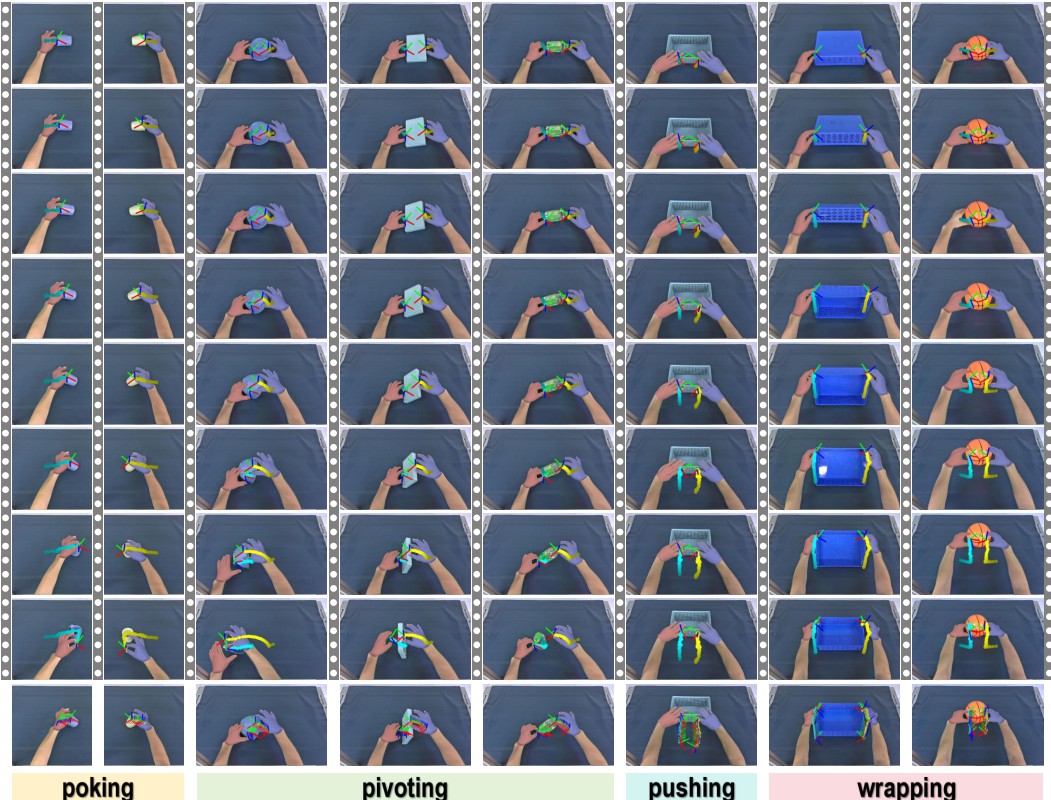

Figure 7: Visualization of extracted hand trajectories from recorded human demonstrations for eight bimanual non-prehensile tasks. Best to view after zooming in.

### A.3 CAPTURE BIMANUAL HAND DEMONSTRATIONS

A major challenge of non-prehensile lies in acquiring complex motion trajectories that involve frequent contact events. To address this, we collect demonstrations directly from bimanual hand motions. For each of the eight tasks across the four skills, a human operator demonstrates the task with one representative object. Following the three-stage BiNoMaP pipeline, these demonstrations are used to extract raw trajectories, which are then optimized and transferred to the dual-arm robot, while parameterized manipulation primitives enable generalization to other instances within the same object category. All demonstrations are recorded using the same platform and camera setup as shown in Fig. 5. Each trial lasts 5 seconds, captured at 10 frames per second, ensuring that both the initial and final hand poses remain within the camera's field of view. From the resulting 50 frames per trial, we manually annotate the start and end frames of each task. Here, Fig. 7 visualizes the extracted human trajectories across the eight tasks, with the bottom row highlighting the effective skill-related segments that serve as the outputs for subsequent trajectory post-optimization.

While human demonstrations provide an accessible means of acquiring complex contact-rich behaviors, they must be carefully constrained to ensure compatibility with the robot's bimanual hardware.

A key reason for adopting human hand demonstrations (rather than relying on teleoperation or simulation) is that existing interfaces cannot reliably produce the fine-grained, millimeter-level coordination required in non-prehensile tasks such as *pivoting a bowl* or *wrapping a basket*. Teleoperation devices introduce latency, lack stable bilateral force feedback, and often produce motions that are smoother or more reflective of the interface rather than the underlying skill. Likewise, simulated demonstrations cannot faithfully capture the nuanced contact formation and frictional transitions crucial to these tasks, and their sim-to-real transfer remains brittle when dense contacts occur.

However, because human hands possess far richer dexterity than two-finger grippers, demonstrations must follow robot-executable motion patterns. During data collection, operators are instructed to *avoid natural but non-transferable strategies*—such as using fingertip rolling, multi-contact regrasping, or in-hand adjustments that the robot cannot reproduce. Instead, the demonstrator performs slow, deliberate, and geometry-consistent motions that mirror the kinematic constraints of the robot's simple grippers. This design choice is essential: allowing more naturalistic human behavior would introduce articulation patterns, micro-adjustments, and compliance effects that cannot be directly mapped to the robot, ultimately degrading the reliability of the extracted trajectories.

To preserve reproducibility and reduce ambiguity in contact formation, the operator also maintains consistent hand-object distances, avoids occluding the object during key transitions, and ensures that contact events follow the same stages defined in our primitive parameterization. These precautions enable the trajectory optimization module to identify stable contact segments and keyframes without requiring force or tactile supervision. Overall, the constrained demonstration protocol strikes a balance between human expressiveness (to capture the essence of each skill) and robot feasibility (to guarantee faithful transfer), which we found to be critical for obtaining reliable manipulation primitives across all four skills.

## B   REPRODUCTION OF ALL SIX BASELINES

**Three Visuomotor Policies**. We reproduced three representative visuomotor policy methods with publicly released code, namely Action Chunking with Transformers (ACT)[3] Zhao et al. (2023a), Diffusion Policy (DP)[4] Chi et al. (2023), and 3D Diffusion Policy (DP3)[5] Ze et al. (2024). For each of these methods, the most critical training data were derived directly from the successful demonstrations obtained through our proposed BiNoMaP pipeline. Specifically, when BiNoMaP successfully executed a given primitive on a target object, we simultaneously recorded the visual observations and the continuous 6-DoF end-effector trajectories of both arms. To ensure that the compared methods converge normally, we collected 50 demonstrations for training on the three skills and six tasks listed in Tab. 1. Compared to conventional teleoperation data collection which is inconvenient to the non-prehensile tasks, these demonstrations offer substantially higher quality and smoother action sequences, since they originate from optimized human-guided trajectories refined via our three-stage framework. We used this dataset to train the visuomotor policies in strict accordance with their respective implementations, including input modalities, network architectures, and action parameterizations. For example, among them, the observation input used by methods ACT and DP is the left and right RGB images collected by the binocular camera, and the input of DP3 is the reconstructed scene-level 3D point cloud. All these policies can be trained on a GeForce RTX 3090 Ti with 24 GB of memory. During evaluation, we applied the trained policies to the same set of non-prehensile tasks to ensure a fair and consistent comparison with BiNoMaP.

To ensure that the reproduced visuomotor baselines were evaluated under fair and meaningful conditions, additional care was taken to remove confounding factors that could degrade their performance. In particular, non-prehensile manipulation is highly sensitive to small perturbations in contact geometry, meaning that inconsistent data quality or action noise can disproportionately affect policy learning. Therefore, we deliberately adopted the cleanest and smoothest possible demonstrations (i.e., trajectories executed by BiNoMaP itself after optimization), rather than using raw human teleoperation data. This choice avoids introducing human-induced jitter, finger–object micro-adjustments, or ambiguous contact transitions that these policies are not designed to interpret. We also aligned the dataset scale (50 demos per task) with or above what the original implementations typically require,

---

[3]https://github.com/tonyzhaozh/act

[4]https://github.com/real-stanford/diffusion_policy

[5]https://github.com/YanjieZe/3D-Diffusion-Policy

and we disabled all optional augmentations that might distort object pose consistency across frames, ensuring that network training remained stable.

Even under these carefully controlled conditions, all three visuomotor policies exhibited substantially lower success rates than BiNoMaP in Tab. 1 and Tab. 4. The performance gap can be attributed to three structural limitations. First, these policies must regress full 6-DoF dual-arm actions at every timestep directly from visual inputs, which amplifies compounding errors when dense contact interactions occur. Second, their training objective optimizes instantaneous action likelihood rather than global contact feasibility, making them prone to drifting away from valid contact manifolds. Third, the policies lack explicit parameterization of skill primitives, making generalization to new object instances considerably weaker. In contrast, our BiNoMaP encodes motion structure through geometric constraints and contact distances. As a result, although the policies were reproduced faithfully and given high-quality data, their performance reflects fundamental differences in representation design rather than implementation bias.

**Three RL-based Sim-to-Real Methods**. We also reproduced three state-of-the-art reinforcement learning approaches that explicitly target non-prehensile manipulation with publicly available code: HACMan[6] Zhou et al. (2023), CORN[7] Cho et al. (2024), and DyWA[8] Lyu et al. (2025). The simulation environment of HACMan is built on top of MuJoCo. While the other two methods CORN and DyWA are built on top of Isaac Gym. Following the training pipelines described in their respective papers and code, we set up simulation environments replicating the physical settings of our selected skills and trained policies with identical object categories and goal configurations. After convergence, the learned policies were transferred to the real-world dual-arm platform using their proposed sim-to-real strategies, including hybrid actor-critic mapping in HACMan, contact representation encoding in CORN, and dynamics-adaptive modeling in DyWA. To ensure reproducibility, we strictly adopted the hyperparameters, reward functions, and training schedules reported in the original works. During evaluation, we tested these distilled student policies under the same experimental settings as BiNoMaP, measuring success rates across multiple trials and skills to assess both task completion performance under the instance-level testing as shown in Tab. 1 (all three baselines) as well as robustness and generalization under category-level distributional variations as shown in Tab. 4 (only the most recent DyWA). Since these RL-based methods need to obtain partial point cloud observations of the target manipulated object when deploying the real world, we utilize VLMs Xiao et al. (2024); Ravi et al. (2025) to detect and segment the instance-level object point cloud $\mathbf{o}_{\text{new}}^{\text{pcd}}$ from the scene point cloud for each testing (the same as in our BiNoMaP method).

For RL-based methods, we prioritized reproducing exactly the experimental conditions described in their papers while avoiding any modifications that might compromise fairness. This included using the officially released simulation assets when available, recreating environment geometries when necessary, and strictly adhering to the original reward shaping, curriculum schedules, action spaces, and termination conditions. When object models differed from those in our evaluation, we recreated category-matched proxies with identical physical properties (mass, friction coefficient, shape primitives) to avoid introducing easier or harder dynamics. Likewise, we disabled domain randomization during training unless explicitly used in the original work, ensuring that the reproduced policies reflect their native sim-to-real assumptions. After training, we transferred all three policies to our hardware platform following their deployment pipelines without applying additional stabilization techniques such as motion smoothing or force filtering, which could bias the comparison.

Despite these efforts, the RL baselines still underperformed BiNoMaP by a large margin in both instance-level (Tab. 1) and category-level evaluations (Tab. 4). This discrepancy arises from structural challenges inherent to RL-based sim-to-real approaches when applied to dense-contact, dual-arm tasks. First, even with high-fidelity physics engines, simulators struggle to faithfully reproduce stick–slip transitions, edge-based pivoting, and compliant wrapping motions, leading to policies that rely on dynamics not present in the real world. Second, these RL methods typically assume single-arm manipulation primitives and therefore require the policy to implicitly discover coordinated dual-arm strategies from scratch, a process far more difficult than BiNoMaP's explicit primitive parameterization. Third, sim-to-real policies depend heavily on single-view point-cloud observations that are inherently noisy in real deployments. Although we apply VLM-based segmentation to match

---

[6]https://github.com/HACMan-2023/HACMan

[7]https://github.com/iMSquared/corn

[8]https://github.com/jiangranlv/DyWA/

their original assumptions, inaccuracies in boundary detection or missing points significantly affect policies that rely on contact-centric reward shaping. In contrast, BiNoMaP bypasses dynamics mismatch altogether by grounding execution in real-world optimized demonstrations, making it inherently more robust to category-level variations and physical deviations.

## C    MORE DETAILS AND RESULTS OF BINOMAP

### C.1    MOVEMENT PATTERNS OF ALL FOUR SKILLS

We first provide further details on how the motion patterns are designed for the four selected non-prehensile skills, as these patterns directly determine how the Geometry-Aware Iterative Contact Adjustment is applied. Illustrative examples of these motion patterns are presented in Fig. 8. For *poking*, since only a single arm is actively involved in manipulation while the other arm remains idle, it is inappropriate to designate the passive arm as the fixed reference. Instead, we select the farthest point from the active end-effector on the object's point cloud as the anchor point, which allows a stable update of Eqn. 4. For *pivoting*, where one arm serves as the pivot and the other performs the motion, we adopt a consistent convention by designating the left arm as the fixed pivot and the right arm as the moving arm. This choice does not compromise the generality of the proposed framework. For *pushing* and *wrapping*, both arms must move simultaneously. In these cases, we arbitrarily choose one arm (by default, the left arm) as the reference frame, while constraining the relative pose and spatial distance of the other arm (the right arm) to remain constant with respect to it. This formulation ensures that Eqn. 4 can be executed robustly.

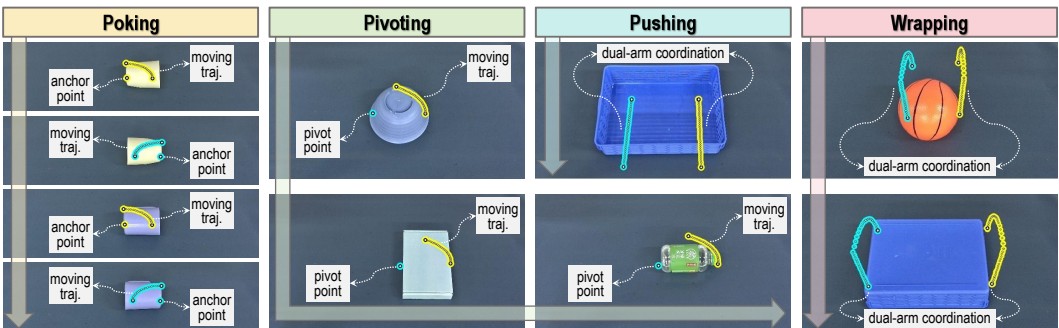

Figure 8: Illustrations of motion patterns for four selected skills. Best to view after zooming in.

### C.2    TRAJECTORY POINTS BEFORE AND AFTER OPTIMIZATION

Next, we explain in detail the dynamic process of trajectory refinement described in Sec. 3.3, specifically corresponding to the second stage in Fig. 2. The Geometry-Aware Iterative Contact Adjustment progressively transforms the raw, coarse trajectories—extracted from human video demonstrations—into smooth and executable robot trajectories. Fig. 9 showcases representative examples of this refinement process for two non-prehensile tasks: *pivoting a bowl* and *wrapping a ball*. In these two cases, one can observe how the originally irregular trajectories evolve into smoother paths with more evenly distributed waypoints. Simultaneously, the corresponding end-effector orientations become either gradually adjusted or consistently maintained. These refined spatial and rotational properties are the key factors underpinning the robustness and stability of the learned bimanual non-prehensile primitives.

### C.3    ADAPTATION AND GENERALIZATION OF MANIPULATION PRIMITIVES

We now provide further insights into how the learned primitives are adapted across different task configurations and generalized across object categories. For instance-level generalization, as discussed in the main paper, we first acquire the point cloud of the object at the demonstration location, denoted as $\mathbf{o}_{\text{base}}^{\text{pcd}}$, using off the shelf VLMs. At test time, we similarly obtain the point cloud of the same object in its new position, denoted as $\mathbf{o}_{\text{new}}^{\text{pcd}}$. By comparing $\mathbf{o}_{\text{new}}^{\text{pcd}}$ and $\mathbf{o}_{\text{base}}^{\text{pcd}}$, we compute the in-plane displacement (e.g., $\Delta x$ and $\Delta y$), and adjust each waypoint in the primitive trajectory accordingly. This mechanism enables effective positional adaptation of the learned skill. For

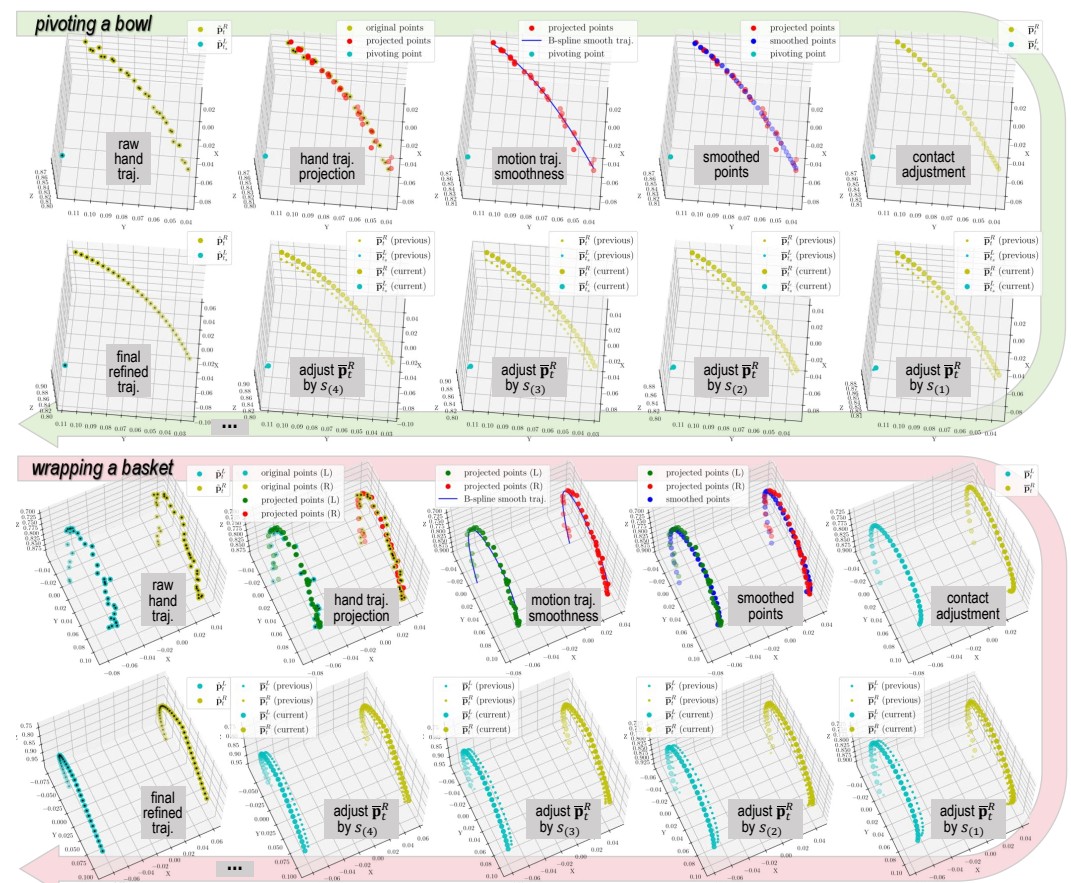

Figure 9: Illustrations of the entire trajectory point optimization process, using skills *pivoting* (top) and *wrapping* (down) as examples. Best to view after zooming in.

category-level generalization, the focus lies on handling shape variations across objects of the same category. As illustrated in Fig. 10, we measure the distance differences between object point clouds along a fixed direction within the horizontal plane of the initial contact points. This approximation provides an estimate of the size discrepancy, which is then incorporated into the trajectory adjustment. For example, in the *pivoting* skill, the moving arm's trajectory is proportionally scaled, while in *pushing* and *wrapping*, the inter-arm distance is increased or decreased synchronously. When both category-level variation and new placements are involved, we combine this scaling procedure with the instance-level adaptation strategy to achieve robust transferability. Together, these two levels of generalization establish the scalability of BiNoMaP, enabling the learned primitives to flexibly adapt across diverse instances and object categories without retraining.

### C.4    More Visualizations of Real Robot Rollouts

Although we have provided snapshots of several key robot actions in real evaluation in Fig. 3, the space constraints prevent us from showing more extensive qualitative results. To address this, Fig. 11 presents additional sequential visualizations of real-world rollouts, highlighting the continuous evolution of object states during execution. These results allow readers to better appreciate the unique characteristics and challenges of non-prehensile bimanual manipulation, particularly its reliance on dense contacts and dynamic stability. For more comprehensive illustrations, we refer readers to our **Supplementary Videos**, which documents the entire execution process in greater detail.

## D    Summary and Analysis of Failure Cases

To better understand the practical performance of BiNoMaP, we collected and summarized representative failure cases observed in real-robot experiments across the four skills and eight tasks, with

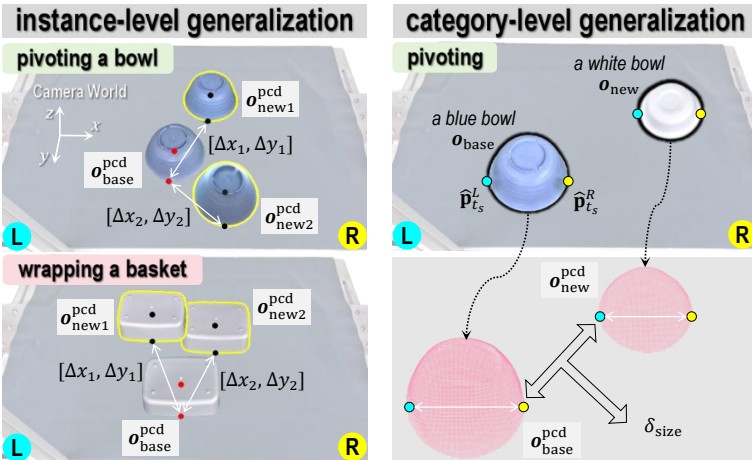

Figure 10: Illustrations for achieving instance-level and category-level generalization of learned primitive manipulation skills. These visualizations serve as a detailed supplement to Sec. 3.4 and the right side of Fig. 2. Best to view after zooming in.

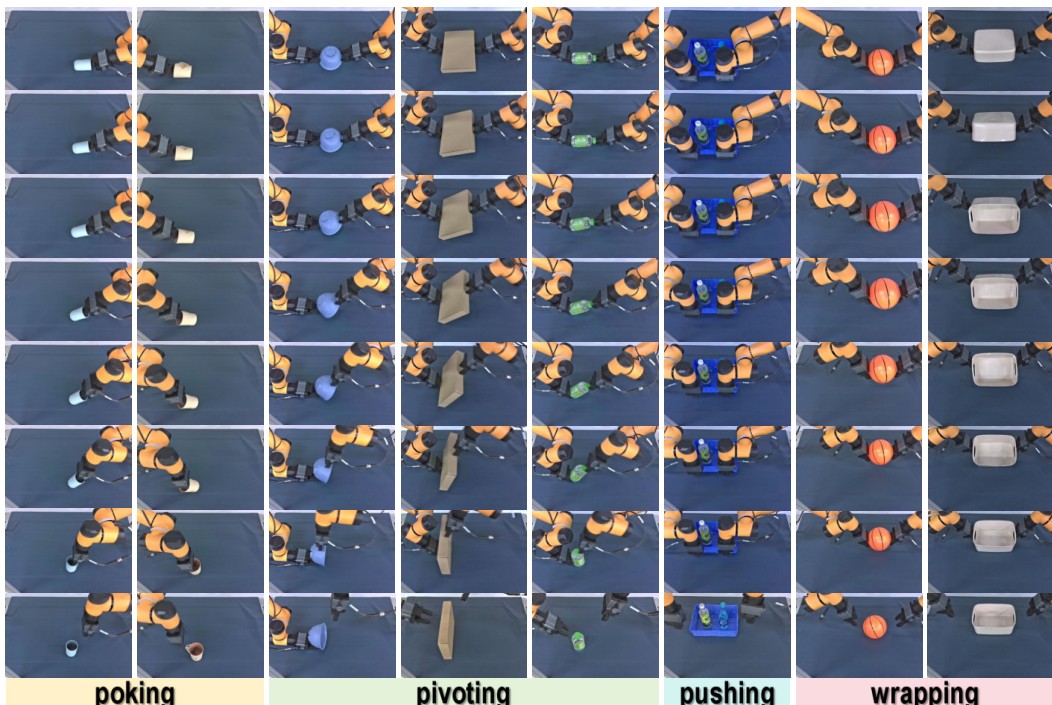

Figure 11: Qualitative real robot rollout samples of all four skills. Best to view after zooming in.

visualizations provided in Fig. 12. We explain the main factors that cause these errors as follow. More additional dynamic details of these failure cases are provided in our **Supplementary Videos**.

- For *poking*, failures occurred when lifting uniformly distributed handless cups due to insufficient insertion of the gripper into the cup mouth, leading to unstable placements and eventual toppling (Fig. 12 (1)(2)), and when handling mugs with uneven mass distribution, where the handle frequently caused rotational displacements under gravity, significantly increasing failure rates due to dynamic and unpredictable changes (Fig. 12 (3)(4)).

- For *pivoting*, failures were observed when flipping bowls, where either insufficient (Fig. 12 (5)) or excessive (Fig. 12 (6)) contact between the right arm's end-effector and the bowl caused slipping or bouncing; when raising thin rectangular boxes, the box either over-tilted toward the left arm (Fig. 12 (7)) and continued falling after release—suggesting the need for corrective post-actions—or was destabilized by excessive end-effector contact (Fig. 12 (8)); and when

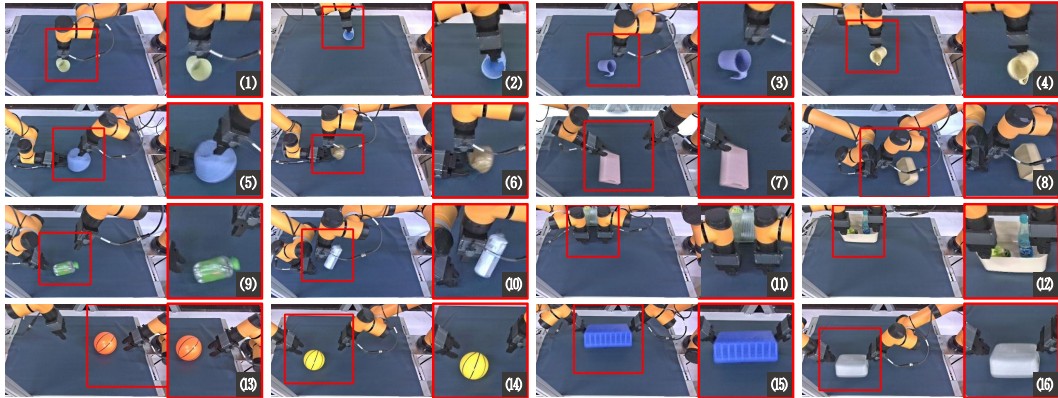

Figure 12: From top to bottom and left to right, we have examples of failed cases in all four skills and eight tasks during real robot evaluation. We have outlined and magnified the areas where the failures occurred so that we can quickly examine them. Best to view after zooming in.

lifting bottles, similar issues of under-contact (Fig. 12 (9)) or over-contact (Fig. 12 (10)) with the bottle's head resulted in slipping or bouncing failures.

- For *pushing*, asynchronous motion between the two arms during basket pushing led to tipping of the heavy objects inside (Fig. 12 (11)(12)).

- For *wrapping*, failures in ball manipulation included arm self-collision (Fig. 12 (13)) and asynchronous release leading to significant moving during placement (Fig. 12 (14)), where the latter is considered an unstable outcome. In the basket-wrapping task, the most frequent failure mode was slippage during lifting and flipping (Fig. 12 (15)(16)), primarily caused by minor discrepancies in the inter-arm distance, which is inherently difficult to maintain perfectly.

**More Hard Non-Prehensile Cases**: As discussed in the final paragraph of the main text, the current BiNoMaP framework, which does not incorporate force–torque sensing, is unable to manipulate objects with extremely low tolerance to errors, particularly in *pivoting* tasks. Typical examples include smooth ceramic or metal bowls, ultra-thin ceramic or metal plates, and fragile glassware, as illustrated in Fig. 13. These rigid objects cannot withstand even visually perceptible deformations, making it challenging to balance contact distance and contact force for successful flipping, even if force sensing were available. We also tested BiNoMaP on these challenging objects, but all of them failed. A potential direction to address this limitation is the use of multi-fingered dexterous hands for more delicate manipulation Li et al. (2025b), or alternatively leveraging the external dexterity such as table edges Wang et al. (2025) to facilitate stable interactions.

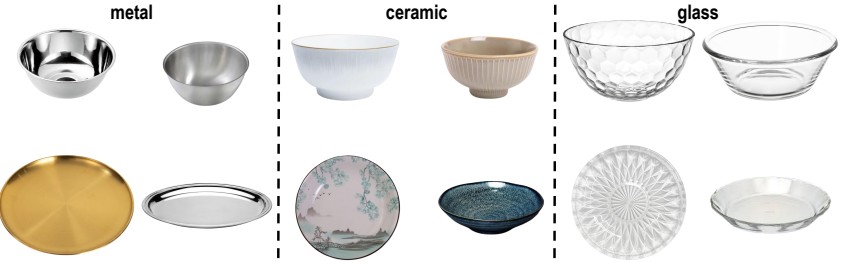

Figure 13: Difficult examples that BioMaP cannot solve at present, especially when these objects are upside down on the table and need to be flipped via applying the bimanual *pivoting* skill.

# E CROSS-EMBODIMENT TRANSFERABILITY OF BINOMAP

To further demonstrate the usability and convenient transferability of BiNoMaP, we evaluate whether the learned manipulation primitives can be directly migrated across different robotic embodiments without re-training. Specifically, as shown in Fig. 14, we validate BiNoMaP on another dual-arm robotic platform configured in a humanoid style. The platform consists of two Rokae xMate CR7[9]

---

[9]https://www.rokae.com/en/product/show/545/xMateCR.html

6-DoF collaborative arms (reach: 988 mm), each equipped with a parallel gripper (Jodell Robotics RG75-300[10], max opening: 75 mm). A binocular camera (Kingfisher R-6000, with specifications identical to those reported in Sec. A.1) is mounted centrally at the head position.

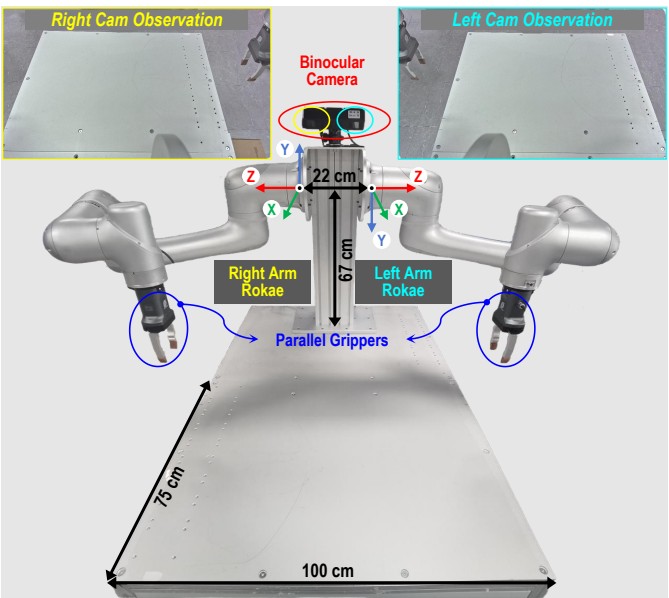

Figure 14: The another dual-arm manipulator platform used for the cross-embodiment evaluation.

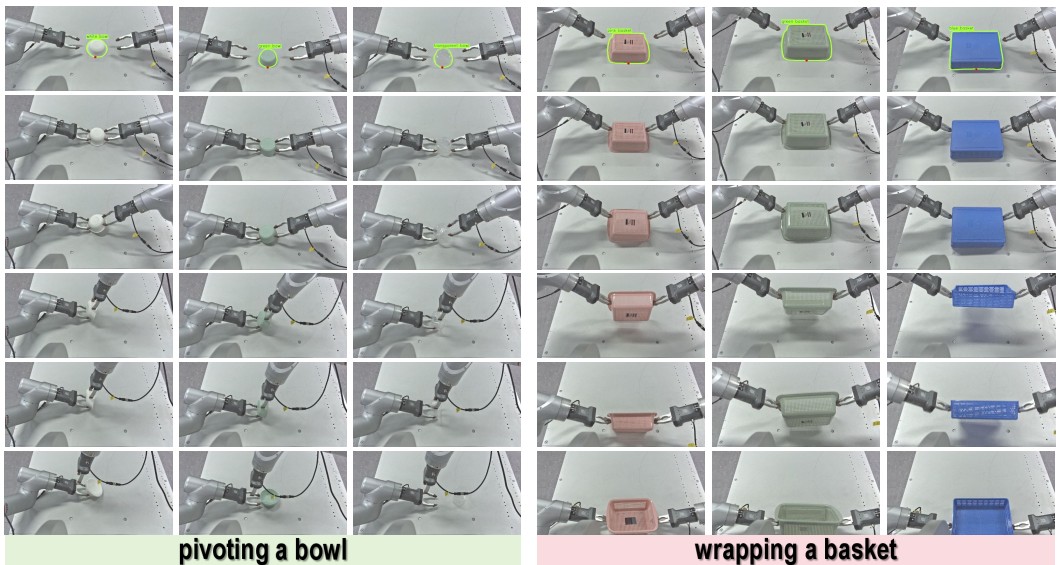

Figure 15: Qualitative real robot rollout samples of two bimanual non-prehensile skills (*pivoting* and *wrapping*) in another dual-arm manipulator platform. Best to view after zooming in.

For this evaluation, we transfer two learned skills (*pivoting a bowl* and *wrapping a basket*), and select three corresponding objects from those shown in Fig. 6 for each task. Aside from the necessary adjustment of the XYZ axis order (due to the humanoid-style mounting differing from the original opposing-arm setup), no significant engineering effort is required. Importantly, unlike prevailing visuomotor diffusion policies Zhao et al. (2023a); Chi et al. (2023); Ze et al. (2024) or RL-based methods Zhou et al. (2023); Cho et al. (2024); Lyu et al. (2025), BiNoMaP does not necessitate recollecting demonstrations or building custom simulation assets for retraining. As illustrated in Fig. 15, the system successfully performs human-like executions of flipping bowls and baskets upright, highlighting the strong cross-embodiment generalizability of BiNoMaP. These results further underscore

---

[10]https://www.jodell-robotics.com/product-detail?id=5

its promising potential for real-world deployment in diverse robotic hardware configurations. Please refer to our **Supplementary Videos** for more dynamic details of these real robot rollouts. Building upon this transferability, we next demonstrate how BiNoMaP can also be seamlessly integrated into more complex downstream applications, showcasing its versatility beyond primitive skill execution.

## F DETAILS OF INTEGRATION WITH DOWNSTREAM APPLICATIONS

This section provides detailed explanations on how the skills learned through BiNoMaP can be integrated to realize three more complex downstream applications. While these demonstrations have already been introduced in the main text (see Fig. 4), here we elaborate on the specific objectives and implementation details of each application.

- (1) The first application demonstrates a **pre-grasping** functionality using the *wrapping basket* task. Two overturned baskets are placed on the table, with their masks accurately identified by VLMs using the prompts "blue basket" and "gray box". The objective is to employ the bimanual wrapping skill to flip the two baskets upright, followed by a bimanual grasping action to lift each graspable large-size basket cooperatively and place it onto the top-left corner of the table. After both operations are completed sequentially, the smaller basket ends up stacked inside the larger one. This application combines two wrapping basket steps with two bimanual grasp-and-place steps (see the left of Fig. 4).

- (2) The second application demonstrates a **rearrangement** functionality using the *pivoting bowl* task. Three overturned bowls are placed on the table, with VLMs accurately locating their masks using the prompts "white bowl", "transparent bowl", and "brown bowl". The objective is to first flip each bowl upright with the bimanual pivoting skill, then use the right-arm gripper to grasp and lift the upright graspable bowls one by one, and place them sequentially onto the top-right corner of the table. After completion, the bowls are stacked in descending order of size from top to bottom. This application combines three pivoting bowl steps with three single-arm grasp-and-place steps (see the top-right of Fig. 4).

- (3) The third application demonstrates an **error recovery** functionality using the *poking mug* task. On the table, a mug is lying sideways toward the right-arm base, alongside an upright uncapped plastic bottle filled with water. Their masks are identified by VLMs with the prompts "blue mug" and "bottle". The objective is first to employ the right-arm poking skill to upright the mug with its handle available for grasping, then grasp the mug handle with the right arm while the left arm grasps the bottle. Subsequently, the two objects are brought together, and the left arm is lifted and rotated to pour water from the bottle into the mug, followed by placing both objects back onto the table. This application integrates one poking mug step, two grasping steps, and two bottle rotation steps (see the bottom-right of Fig. 4).

For the entire dynamic process details of these applications with mixed prehensile manipulation and non-prehensile manipulation, please refer to the **Supplementary Videos** we provide. Overall, these three downstream examples serve as preliminary demonstrations of the application potential of bimanual non-prehensile manipulation. We envision that combining such BiNoMaP skills with multimodal large vision–language models can enable more complex, long-horizon, and interactive robot manipulation in the future.

## G FURTHER VERIFICATION OF GENERALIZATION AND ADAPTABILITY

### G.1 ADDITIONAL CROSS-OBJECT SKILL TRANSFER EXPERIMENTS

To further substantiate the generalization capability of BiNoMaP, we extend the set of cross-object skill transfer experiments beyond those included in the main paper. As previously demonstrated, BiNoMaP is not only effective for objects with substantial intra-class geometric variation but can also transfer skills across heterogeneous object categories with drastically different shapes. For the *pivoting* skill, our original experiments already covered three geometrically distinct objects: a bowl (hemisphere), box (cuboid), and bottle (cylinder). For the *wrapping* skill, we demonstrated transfer across a ball (sphere) and basket (cuboid).

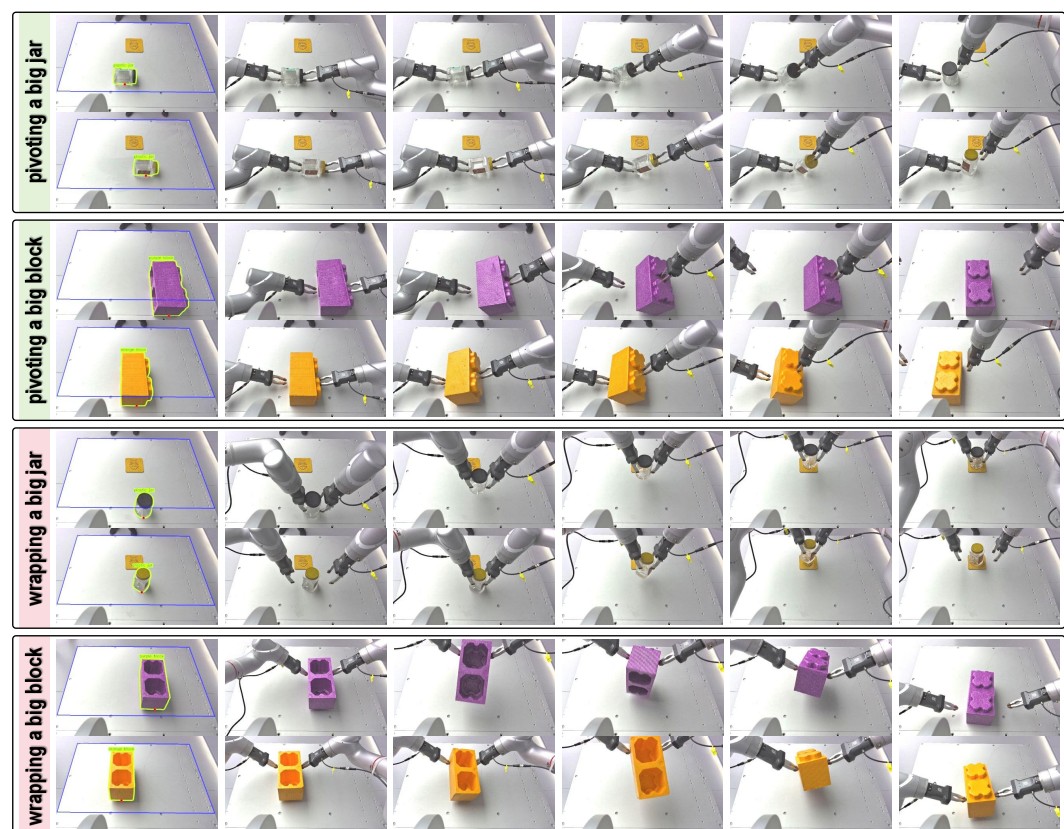

Figure 16: Visualization results of new skill transfer experiments performed on the humanoid dual-arm robot. We demonstrated that BiNoMaP can be applied to new objects (*e.g.*, the big jars and blocks) with shapes completely different from assets we used before. Best to view after zooming in.

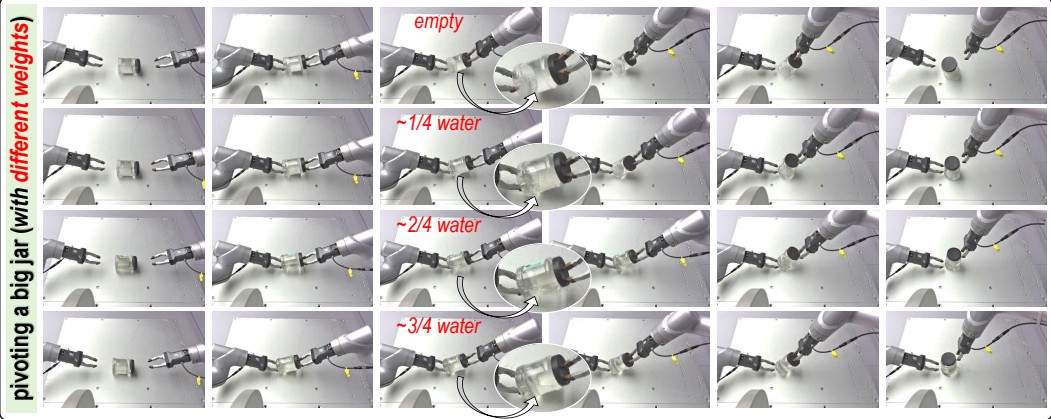

Figure 17: Visualization of experimental results regarding the impact of object mass on the success rate of non-prehensile manipulation. We filled the big jar with different weights of water to perform reasonable and controlled tests. Best to view after zooming in.

To further highlight BiNoMaP's practical usability and strong geometric generalization, we conducted additional real-world experiments on the new humanoid dual-arm platform (refer Fig. 14). Specifically: (1) The *pivoting* skill was transferred to two new objects with significant shape differences: a big jar and a big block. The jar was tested both empty and filled with water of varying mass, demonstrating that BiNoMaP remains robust under different mass distributions. (2) The *wrapping* skill was similarly transferred to the big jar and big block, expanding beyond the ball and basket used in the original evaluation.

Representative visualizations of these new trials are provided in Fig. 16 and Fig. 17. And the corresponding full dynamic rollouts are included in the attached **Supplementary Videos**. We believe that these new experiments further validate BiNoMaP's generalization when there have significant variations in shape, size, or mass distribution of new unseen objects.

## G.2 DENSE SPATIAL GENERALIZATION TEST ACROSS OBJECT PLACEMENTS

We further validated BiNoMaP's adaptability to spatial variations through a dense placement test conducted on the new humanoid dual-arm platform. Focusing on the *pivoting a bowl* task, we sampled a wide range of feasible object positions within the dual-arm robot's reachable workspace. In total, 22 distinct locations were evaluated, and 34 rollouts were executed, achieving a success rate of approximately 64.7% (22/34). This dense sampling study demonstrates that BiNoMaP maintains stable performance under substantial spatial perturbations without requiring retraining or additional optimization steps. Visualization of the spatial layout and execution results is provided in Fig. 18, with corresponding dynamic videos included in our **Supplementary Videos**.

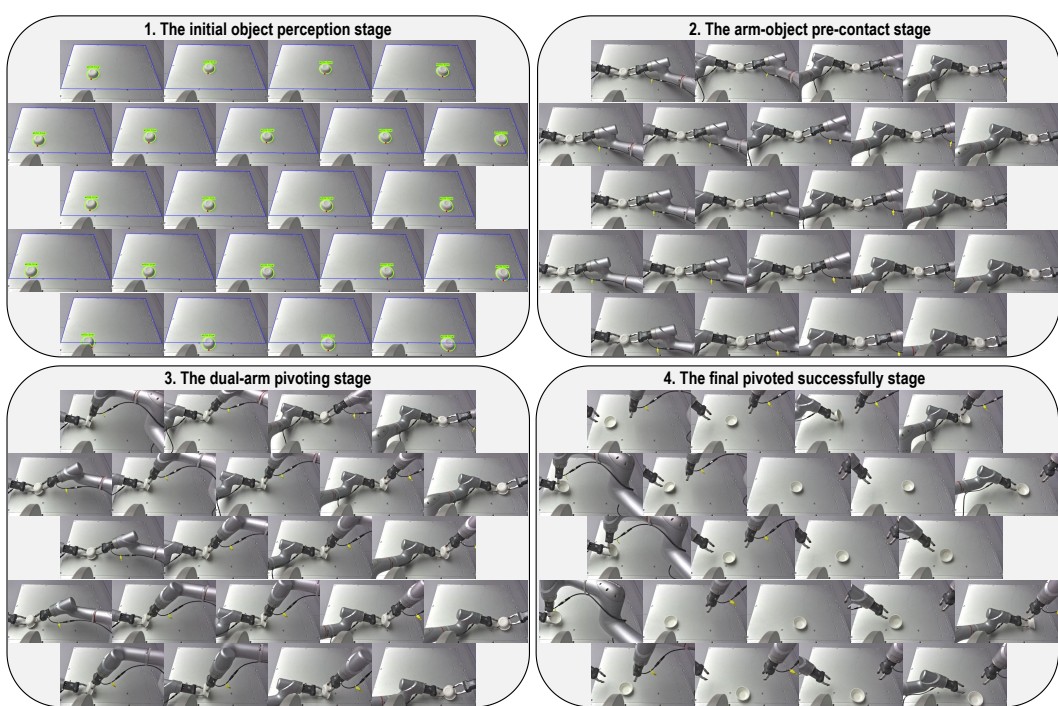

Figure 18: Visualization of qualitative test results for the *pivoting a bowl* task. We have conducted and passed extensive testing at 22 different initial locations. The illustrated frames include segmentation of the target object, initial pre-contact, dual-arm coordinated pivoting, and a screenshot of the final task completion. Best to view after zooming in.

## H STATEMENT ON THE USE OF LARGE LANGUAGE MODELS

In the preparation of this manuscript, we employed the ChatGPT large language model solely for the purpose of **language refinement and stylistic polishing**. Specifically, the model was used to improve the clarity, readability, and conciseness of our writing. Importantly, no part of the research design, methodology, experimental setup, or technical innovation was generated or influenced by the model. All conceptual developments, algorithmic designs, and empirical analyses were conceived and executed entirely by the authors. The use of ChatGPT was strictly limited to assisting with the presentation of our work, and we ensured that the scientific contributions and intellectual content of the paper remain fully attributable to the authors.

