# OpenReview forum: "BiNoMaP: Learning Category-Level Bimanual Non-Prehensile Manipulation Primitives"
_ICLR.cc/2026/Conference — Submitted to ICLR 2026_

### Official Review · Reviewer_VeaC · 2025-10-23

**Soundness:** 3
**Presentation:** 3
**Contribution:** 3
**Rating:** 8
**Confidence:** 4

**Summary:**

The paper presents a visuomotor method to perform several tasks of non-prehensile manipulation using a dual-arm robotic platform and a camera system able to reconstruct 3D motion.

The work employs human demonstration to collect human-inspired movement strategies to perform the given manipulation tasks, and then - by means of their proposed method - it transfers these movements to the robots showing satisfactory performance across various tasks both seen and unseen, with a good degree of generalization.

The paper is benchmarked against state-of-the-art methods showing superior performance. Moreover, the robustness of the proposed method is assessed by different ablation studies, showing a good degree of resistance to parameters' changes.

**Strengths:**

1. The paper is original in its way of using demonstrative manipulation data and robustly generalize them to robot movements beyond the initial dataset.
2. The paper overall quality is high and can be appreciated by the level of depth in which each aspect of the proposed method has been evaluated.
3. The paper is clearly presented and easy to follow.
4. The paper most significant result is - in my opinion - the ability to generalize the learned movements beyond the initial training demonstrations.

**Weaknesses:**

1. The idea of decomposing movements in atomic/primitive actions is not new and fairly renowned in robotics. I recommend reviewing the seminal works on: Dynamic Movement Primitives (Schaal), Dynamic Motor Primitives (Hogan and Sternad), Elementary Dynamic Actions (Nah), Large Behavioral Models (Tedrake).
2. The learning from human demonstration is a very reasonable approach. However, in your work I had the impression that the human demonstrators were not using their most natural way to perform the task. For example, the pivoting task of cups and mugs will be normally performed in a single arm movement by grasping the object, rather than by using a bi-manual non prehensile strategy. The same applies for many other movements. What instructions did you give to the subjects? If the movements were so structured, wouldn't have been more reasonable to directly learn from robotic demonstration?
3. The presentation of the six baseline methods could be improved, especially in the presentation of the way such methods are implemented in your platform. I find the superior performance of your methods both impressive and worrisome. Is there a way for others to confirm your results in any way?

**Questions:**

In the following you can find a detailed list of all the questions I would like for you to address.

1. In the abstract the authors claim that "pushing, poking, and pivoting" are contact-rich actions. However, I would argue quite the opposite. Most of the time, humans push or poke thing with either the palm of their hand or a finger. Pivoting is maybe more complex from a manipulation standpoint, but it is hardly contact-rich. I understand the works focuses on non-prehensile manipulation, but I struggle to see how this could be contact rich. Maybe either consider re-framing the sentence or better explaining what you mean with contact rich.

2. The authors wrote: "To the best of our knowledge, our work is the first to propose a fully RL-free paradigm for learning bimanual non-prehensile manipulation skills through imitation and geometric reasoning.". I invite authors to consider the paper: https://ieeexplore.ieee.org/abstract/document/10801502. While this work is not directly applied to non-prehensile manipulation, it leverages learning by demonstration and geometric reasoning to develop modular robotic primitives of motion. It would be worth including it in your paper and discuss your position with respect to it.

3. In the bimanual robotic manipulation section, I found no mention of the recent efforts and convincing results obtained by Toyota Research Institute on bimanual manipulation using large behavioral models and diffusion policies.
I strongly invite authors to consider their work and expand their evaluation of the state-of-the-art. Here you can find the link to the most recent YouTube video showcasing the performance of these large behavioral models: https://www.youtube.com/watch?v=DeLpnTgzJT4 .

4. While I appreciate the efforts in making Figure 2, I also have to admit that the figure is very dense of information and hard to navigate. Is there a way to simplify or improve the Figure readability? This is more of a minor comment, but it would definitely benefit the overall quality of the work.

5. In Section 3.3 the authors claim that a prerequisite for all skills is coplanarity of trajectory points in each arm. Why is that? It's this a mathematical need to simplify computation or a physical need that would make manipulation impossible otherwise? I think it would be important to clarify better this point.

6. Line 288-289: there is a typo, the authors wrote "safty" but I believe they meant "safety".

7. The authors claim that the contact adjustment takes less than 5 minutes. While this is not a long time, it is also not very fast. Is there a better way to improve contact detection? Maybe briefly discuss this point later in the paper.

8. I agree that your geometry-aware optimization implicitly encodes object dimensions, but what about object physical properties such as stiffness, inertia or damping? What will happen if you try to manipulate an object of the same dimensions but very different compliance - for example much softer/stiffer or much heavier/lighter? Please comment on this in the paper.

9. A minor point: sometimes in the text I found a different type of font. Can you please uniform the type of font across the paper?

10. Table 1 results are very promising. What is your explanation behind your method much better performance? Could it also be that you implemented those other methods in an experimental setup which is not ideal for them?

11. Line 368-369: there might be another typo, the authors wrote "Details of reproduced all baselines are ... " but I believe they meant "Details of all reproduced baselines are ...".

12. Regarding research answer (A1), I think the authors should spend considerably more efforts in explaining how they trained the other benchmark methods. While I don't doubt the results of their proposed methods, I am surprised by the poor performance of the other methods. How is that possible? Can you provide - even in the supplementary material - sufficient and QUANTITATIVE justifications on why the alternative methods are so underperforming? The same observations applied to answer (A3).

13. Did category level generalization also include substantial variations in object stiffness and weight? I think these are extremely important factors, since the ability to maintain contact with object will be heavily influenced by both the object rigidity and its weight.

14. I'm not really sure what is the semantic difference between 'poking' and 'pivoting'. In both cases you are 'flipping' the object to its correct orientation. Why did you separate them? Is it because of the different strategy adopted by the humans? Also, I do expect human demonstrator to put a cup or mug in a vertical position by using just one-hand with a prehensile grasping movement. I have rarely see humans putting a cup upright by poking inside of it.

15. Related to the previous point, what instructions did you give to the human demonstrators? Did you 'bias' them towards a specific strategy that you knew would better translate to robot movements? It's totally fine if you did so, but you need to specify it. If I look at Figure 7, I notice that the poking task is performed in a very 'unnatural' way for a human, the same goes for wrapping the basketball. Most humans will not lift a basketball in that way. Similar observations could be done also for the pivoting and pushing task.
In other words, if you want to learn by demonstration but will instruct users to move in specific ways, why not directly teleoperating the robots to do so and learn from the robotic teleoperation? What is the need of the human at this point?

16. Section A.3 of the Appendix needs to be improved to answer my previous questions. As of now, I have the impression that the human demonstrations are not needed, since humans are not performing the tasks in their natural way, thus making learning from human demonstration meaningless in my view. This does not invalidate your methods, it just makes the human demonstration easily replaceable with robotic demonstration via teleoperation. Don't you agree? If not, please comment extensively, I'll be happy to change my mind.

17. In Section B of the Appendix you discuss how you performed the reproduction of the six baseline methods. I think it would be beneficial to extend this section to better discuss any limitation of your training methods for these baseline techniques. I ask this specifically because your method seems to overperform all the baseline by a very large amount, and I want readers to be able to verify your claims as closely as possible. Also, did you release the code for YOUR method as well as the code for YOUR implementation of the baseline methods? If not, I strongly recommend to do so.

18. A minor point on the video. The video is well organized and clear but it's very fast. I often had to pause the video and go back and forth to well appreciate the results. I recommend slowing the whole video by a factor of 2 or 3.

---

> ### Author Response · Authors · 2025-11-24
> **Response by Authors to Reviewer VeaC [Part 1]**
>
> ## Overall Response
>
> We thank `Reviewer VeaC` for the careful reading and for the strong endorsement of our work — in particular the recognition of (1) the novelty of leveraging human demonstrations to obtain robust, transferable bimanual motion strategies, (2) the thoroughness of our empirical evaluation, and (3) the demonstrated ability to generalize learned movements beyond the original demonstrations. We greatly appreciate the detailed, constructive list of issues and suggestions .Those will help us improve both the manuscript and the supplementary material.
>
> Briefly, we emphasize the main practical and scientific advances of BiNoMaP that are most relevant to the reviewe's remarks: (1) a truly **RL-free, sim-free** pipeline that extracts bimanual primitives from human video and directly targets real dual-arm hardware; (2) a **coarse-to-fine trajectory extraction** and **geometry-aware post-optimization** procedure that resolves perceptual and embodiment gaps to produce robot-feasible, contact-aware motions; and (3) a **parameterized primitive representation** enabling both instance-level and category-level transfer, validated through extensive real-robot experiments, ablations, and additional rebuttal-phase tests (including shape transfers and dense spatial sampling).
>
> The reviewer raises many valuable technical and presentation points (e.g., connections to prior DMP-like literature, demonstration protocol, baseline reproduction details, robustness to physical properties, and minor editorial fixes). Below we will address each item in turn with concrete clarifications, additional experiments, and improved manuscript text.
>
> ***
>
> ## (A) Response to Weakness 1: On the novelty of primitive decomposition
>
> We thank the reviewer for highlighting this point. Indeed, the general idea of decomposing behaviors into motion primitives has a long history in robotics (e.g., DMPs, EDAs, LBM). Our work does not claim conceptual novelty in proposing *primitive-based decomposition per skill*. Instead, BiNoMaP focuses on **learning highly task-specific, contact-aware, and short-horizon non-prehensile bimanual motion sequences**, each of which lasting only 3–5 seconds and typically forming an indivisible continuous start-contact–to–end-contact action. In this regime, further decomposition offers limited benefit, whereas combining these primitives into long-horizon tasks (as illustrated by the three downstream tasks in `Fig. 4`) is where hierarchical structure becomes essential.
>
> We fully agree that positioning our method within this literature improves clarity. Following the reviewer's suggestion, we have added a new sentence in `Section 1 fourth paragraph` (highlighted in red in the revised manuscript) that discusses the seminal works recommended by the reviewer and clarifies how BiNoMaP differs in scope, granularity, and execution setting.
>
> ## (B) Response to Weakness 2: On human demonstration strategies and their deviation from "natural" human behavior
>
> We appreciate this insightful observation. It is true that our human demonstrators did not perform the tasks in the most natural human manner. For example, using single-hand grasping to upright a cup or simply lifting an inverted bowl. However, these natural strategies are **not executable on current dual-arm robotic platforms**, for several practical reasons:
>
> > (1) **Lack of tactile sensing and insufficient friction** at robot end-effectors make prehensile grasping of thin rims or slippery surfaces unreliable.
>
> > (2) **Risk of collision** between parallel-jaw grippers and the table in upright-cup or inverted-bowl scenarios.
>
> > (3) **Compliance and fragility of household objects** make grasp-and-lift strategies potentially damaging when performed without force sensing.
>
> > (4) Even dexterous multi-fingered hands struggle with these tasks due to **limited tactile resolution**, **limited finger friction**, and **insufficient dexterity** compared to humans.
>
> Therefore, we intentionally **instructed human demonstrators to follow strategies that are feasible, safe, and reliable for our bimanual robot**, rather than their natural human routines. This design ensures that the extracted trajectories are directly transferable to hardware.
>
> Regarding why we did not collect robot teleoperation demonstrations instead: **obtaining high-quality, contact-rich, bimanual non-prehensile demonstrations via teleoperation is extremely difficult**. We attempted this on two platforms (a dual-arm Aloha system and a VR-teleoperated humanoid robot), and all trials of bimanual non-prehensile skills (e.g., pivoting a bowl or bottle) failed. We include the corresponding evidence in the supplementary video (`new_exps_teleop_test.mp4`). Hence, human demonstrations, which are performed in a robot-executable manner, were the only viable source for collecting such skills in real world.

---

> ### Author Response · Authors · 2025-11-24
> **Response by Authors to Reviewer VeaC [Part 2]**
>
> ## (C) Response to Weakness 3: On baseline selection, reproduction, and fairness of comparison
>
> We fully acknowledge that benchmarking a simulation-free, RL-free real-world method against existing approaches is nontrivial. To ensure fairness, we selected:
>
> > (1) **Three RL-based non-prehensile manipulation methods** specifically designed for similar tasks, and
>
> > (2) **Three widely adopted visuomotor imitation-learning baselines**, which, while mostly evaluated on grasping tasks, include limited non-prehensile capabilities (e.g., pushing-T, planar spreading).
>
> Reproducing these baselines in our setting presents genuine challenges:
>
> > (1) RL-based methods require **adaptation to our task-specific object set**, including importing simplified meshes, tuning contact/dynamic parameters, and determining how to extend their originally single-arm policies to bimanual settings. In practice, we adopted the most conservative synchronous/asynchronous variants without altering any core learning modules.
>
> > (2) For imitation-learning baselines, **robot teleoperation data cannot be collected** for our contact-rich non-prehensile tasks, so the only feasible option was to use BiNoMaP's successful rollouts as training data. Consequently, it is expected that these baselines do not outperform BiNoMaP. Since a student model rarely exceeds the performance of its teacher.
>
> We agree that additional clarification is helpful. Therefore, in the revised supplementary material, we have expanded the baseline reproduction section with detailed discussions on adaptation strategies, implementation constraints, and fair-use considerations.
>
> ***
>
> ## Q1. On using the term contact-rich non-prehensile manipulation
>
> We appreciate the question. In this work, we follow the long-standing convention in non-prehensile manipulation literature, where such skills are typically regarded as contact-rich because they rely on sustained non-enclosing contact between the end-effector and the object. Similar to prehensile "clamped/unclamped" transitions, most non-prehensile primitives naturally start and terminate with contact making/breaking. Thus, using this terminology is consistent with prior usage and should not introduce conceptual ambiguity. We agree a more precise term might emerge in future literature.
>
> ## Q2. On potential overclaim in the introduction
>
> Thank you for highlighting this. We have removed this overclaiming sentence in the revised draft. Following your suggestion (also discussed in `Weakness #1`), we added a discussion of seminal modular robotic primitives in `Section 1 fourth paragraph` (highlighted in red). We expect that this helps to improve completeness without affecting the paper's core claims.
>
> ## Q3. On missing discussion of TRI's large behavioral models for bimanual manipulation
>
> Thank you for pointing out this highly relevant work. We are familiar with TRI's strong contributions, including the popular diffusion policy, and agree the LBMs line of work should be discussed. Our initial focus was on non-prehensile skills, and we inadvertently overlooked this important bimanual literature. The revised manuscript now includes this discussion in `Related Work` (highlighted in red).
>
> ## Q4. On readability issues in Fig. 2
>
> We appreciate this observation. `Fig. 2` attempts to present the complete three-stage BiNoMaP pipeline, which inevitably increases visual density. We retained full-resolution assets so readers can zoom in without losing detail. After de-anonymization, we will release all figure source files and high-resolution elements on our project webpage to further improve clarity and transparency.
>
> ## Q5. On the coplanar assumption in trajectory optimization
>
> Each primitive's trajectory is a sequence of 6-DoF waypoints. The coplanar assumption applies only to the **3-DoF positional component**, enabling the planar constraint optimization in `Eq. (2)`. The orientations (remaining 3-DoF) are unconstrained. All four non-prehensile primitives evaluated in the paper naturally satisfy this coplanarity, which stabilizes optimization, reduces jitter, and lowers the risk of arm–object detachment during execution. This assumption is therefore broadly valid without harming skill expressiveness.
>
> ## Q6. Typographical error
>
> Thank you for catching this. The typo is fixed in the updated manuscript.
>
> ## Q7. On the time cost of coarse-to-fine trajectory optimization
>
> Thank you for raising this. The 3–5 minute optimization cost refers only to the one-time real-world coarse-to-fine refinement needed to determine a new object's optimal contact distance. Once optimized, this distance is reused for subsequent executions of that object (or its identical copies), so runtime per skill becomes very short. We clarified this in the revised draft (highlighted in red).

---

> ### Author Response · Authors · 2025-11-24
> **Response by Authors to Reviewer VeaC [Part 3]**
>
> ## Q8. On material properties, object mass, geometry variability, and robustness
>
> Unlike RL-based simulation methods that require explicit parameters (stiffness, inertia, damping), BiNoMaP learns directly from real demonstrations and feedback, making it inherently robust to material and physical variations. This is evidenced by our experiments on ~50 diverse objects shown in `Fig. 6` (paper, plastic, ceramic, leather, etc.), achieving ~76% average success rate. However, because our platform lacks force sensing, BiNoMaP cannot handle objects that are rigid, fragile, or allow no deformation (e.g., upside-down rigid bowls/plates, as discussed in `Fig. 13`).
>
> To address your question on shape and mass variations, we added new results in the `Appendix Figs. 16–17` and corresponding videos (`new_exps_jar_block.mp4`), showing BiNoMaP successfully transferring to distinct geometries and masses.
>
> ## Q9. On font usage for naming the four primitives
>
> You are correct that the typewriter font (\texttt) was used only for emphasizing the four primitives (poking/pivoting/pushing/wrapping). To maintain consistency and avoid distracting formatting, we have unified the fonts and removed the \texttt usage in the revised manuscript.
>
> ## Q10. On fairness of baseline reproduction and large performance gaps in Table 1
>
> We understand the concern. This question is fundamentally aligned with `Weakness #3`. To ensure a fair and objective comparison in `Table 1`, we had to carefully adapt and reconstruct all six baselines (often requiring non-trivial modifications). For RL-based methods, we used simple geometric meshes for simulation import, and applied only the minimal asynchronous/synchronous extensions needed to enable bimanual execution, avoiding any sophisticated redesigns. Because we could not obtain teleoperated demonstrations required by visuomotor diffusion policies, we used BiNoMaP's successful rollouts as training data. Otherwise a meaningful comparison would have been impossible. Consequently, their performance being lower than BiNoMaP is understandable, as such methods typically cannot surpass the "teacher model" that generated their data.
>
> Although stronger hyperparameter tuning or more intensive system engineering might improve their performance, pursuing algorithmic innovation in these baseline paradigms is outside our scope. Importantly, BiNoMaP's advantages have limits: `Table 4` shows performance drops on unseen objects, and `Fig. 13` shows non-feasible cases. These outcomes highlight the inherent difficulty of real-world non-prehensile manipulation, rather than suggesting that BiNoMaP solves the problem completely.
>
> ## Q11. On the grammatical error
>
> Thank you for identifying this mistake. We have corrected it in the updated manuscript.
>
> ## Q12. On concerns that all baseline methods show poor performance
>
> We fully understand the concern. The performance gap mainly stems from the **fundamentally different technical paradigms** represented by the baselines comparing to BiNoMaP:
>
> > (1) **RL-based approaches** are typically optimized in simulation for a narrow set of geometric objects. Sim2real transfer is known to be fragile even without changing task or object. For example, DyWA reports ~68% real-world success on single-arm poking. Our reproduced results (21/30 and 19/30 for poking cup/mug) closely match its original performance. However, extending such RL methods to bimanual pivoting or wrapping introduces substantial complexity: continuous contact, delicate arm–object distance control, and strict inter-arm coordination. This dramatically increases exploration difficulty and leads to sharp performance drops.
>
> > (2) **Visuomotor imitation learning** suffers from data scarcity. Since real demonstrations could not be collected for all objects, we used BiNoMaP rollouts, which limits the achievable performance.
>
> Thus, the poor baseline performance does not indicate unfairness, but reflects the **mismatch between existing paradigms and the complexity of bimanual non-prehensile tasks**. We hope this highlights the need for new approaches tailored to real-world contact-rich manipulation.
>
> ## Q13. On sensitivity to stiffness and weight variation
>
> This is indeed an important question for contact-rich manipulation. As partially addressed in `Q8`, BiNoMaP requires the end-effectors to apply inward pressure to create stable contact and reaction forces. For highly rigid, low-friction objects (e.g., metal/ceramic/ glass bowls in `Fig. 13`), the grippers—without force sensing and with similarly low friction—cannot generate sufficient force balance, making such objects currently infeasible.
>
> In contrast, BiNoMaP is robust to weight variation: `Fig. 17` shows successful pivoting of a big jar containing different water levels. The corresponding video (`new_exps_jar_block.mp4`) are provided in the supplementary materials.

---

> ### Author Response · Authors · 2025-11-24
> **Response by Authors to Reviewer VeaC [Part 4]**
>
> ## Q14. On the intrinsic differences between skills
>
> Thank you for raising this essential conceptual question. The differences between skills do not originate from human demonstrators (see our response to `Weakness #2`), but from the inherent structure of each skill when instantiated on specific objects—i.e., **skill + object = task**. For example:
>
> > (1) **Poking** relies on side-surface contact along the long axis of the gripper. The object must provide a concave region enabling partial insertion, and the opposite side must create a stable stand-point on the table.
>
> > (2) **Pivoting** relies on tip contact with slight over-pressing, requires the tips not to slip during lifting, and demands blocking the opposite side of the object, which is a role fulfilled by the second arm rather than a wall (as in prior work).
>
> These distinctions are formally defined in `Appendix A.2`. We hope this clarifies the structural differences between the skills.
>
> ## Q15. On the necessity of human bimanual demonstrations
>
> This is a valuable point. As discussed earlier (`Weakness #2` and `Q14`), real-world bimanual non-prehensile manipulation lacks accessible sources of full-state demonstrations, and conventional teleoperation hardware cannot produce the required fine-grained motions (see the video `new_exps_teleop_test.mp4`). We intentionally avoid RL-based sim-to-real pipelines because of their brittle transfer behavior. Therefore, leveraging human demonstrations is the most practical and efficient choice.
>
> Since human hands differ from two-finger grippers, demonstrations must follow gripper-executable motion patterns—slow, simplified, and non-dexterous. Moreover, raw human trajectories alone are insufficient, hence the importance of our post-processing and optimization modules. These components are indispensable for enabling robots to replicate the demonstrated skills. In future work, we plan to upgrade to multi-fingered end-effectors, which will allow more natural human-style demonstrations.
>
> ## Q16. On whether the human demonstration module is essential
>
> Thank you for emphasizing this. As clarified in `Q15`, the human demonstration module is essential and cannot be removed or trivially replaced. We agree with the reviewer that these discussions should be included in the appendix, and they have been added to `Appendix A.3` in the revised manuscript.
>
> ## Q17. On including more implementation details and releasing code
>
> Thank you for the suggestion. As addressed in `Weakness #3` and `Q10`, we have added more details on the reproduction of baselines in `Appendix B` (highlighted in red) to improve clarity and reproducibility. Regarding BiNoMaP, we plan to release the full implementation after de-anonymization to support the community's progress in bimanual non-prehensile manipulation.
>
> ## Q18. On supplementary video formatting and playback speed
>
> Thank you for this helpful feedback. Due to the 100MB size limit, we had to compile many rollout videos into PPT-based grouped segments, then record and export them as a single video file. Uploading all raw videos separately was infeasible and would have required reviewers to open dozens of individual files. Most clips were sped up 2× to stay within the time limit. Reviewing them at **0.5× playback speed** will reproduce real-time and more comfortable motion. After de-anonymization, we will upload all raw videos individually on the project website to ensure convenient and detailed inspection. We apologize for the inconvenience and appreciate your understanding.

---

### Official Review · Reviewer_AWJr · 2025-10-31

**Soundness:** 3
**Presentation:** 3
**Contribution:** 2
**Rating:** 4
**Confidence:** 5

**Summary:**

This paper introduces a novel approach to non-prehensile manipulation by making two key contributions. First, it moves beyond single-arm systems reliant on environmental aids, proposing instead a generalizable dual-arm configuration and a corresponding suite of Bimanual Non-prehensile Manipulation Primitives (BiNoMaP). Second, it abandons the prevailing RL-based paradigm for a three-stage, RL-free framework. This method learns from human video demonstrations, refines the extracted trajectories through a geometry-aware optimization to bridge the visual-morphological gap, and finally parameterizes them with object attributes for category-level generalization.

**Strengths:**

1. The paper is well-written, with a clear and logical structure.
2. The paper provides extensive experiments to demonstrate the effectiveness of the proposed method.
3. This paper offers an analysis of failure cases and discusses how integration with existing Vision-Language Models (VLMs) could enable more advanced robotic behaviors.

**Weaknesses:**

1. The claim that the post-optimization method is more effective and safer than simulation or real-world RL requires experimental validation. A comparative study is needed to substantiate this advantage.
2. The category-level generalization appears limited. The demonstrated objects within each category primarily vary in size but not in shape, whereas prior works often generalize across objects with more significant morphological differences.
3. An analysis of the method's robustness to inaccurate object size estimation is necessary. My understanding is that the size-based generalization is highly dependent on accurate size measurement. In real-world scenarios, however, obtaining precise object dimensions from point clouds is often unreliable.
4. I would like to see an evaluation of the method's generalization capability with respect to variations in object position.
5. For a paper relying solely on real-world experiments, 10 test trials seem insufficient and may introduce randomness, potentially failing to accurately reflect the method's true performance.

I will consider raising the score if the rebuttal of the author can address the above concerns.

**Questions:**

please see the weakness.

**Details Of Ethics Concerns:**

no ethics concerns

---

> ### Author Response · Authors · 2025-11-24
> **Response by Authors to Reviewer AWJr [Part 1]**
>
> ## Overall Response
>
> We thank `Reviewer AWJr` for the careful reading and constructive feedback, and we appreciate the positive recognition of our paper's clarity, extensive experiments, and the failure-mode analysis that connects BiNoMaP to vision–language integration. We are encouraged that the reviewer finds the problem setting and experimental effort compelling.
>
> We briefly emphasize the practical advantages of BiNoMaP that particularly relate to the reviewer's comments: (1) an **RL-free, sim-free** pipeline that reduces engineering and training overhead; (2) a **trajectory-extraction + geometry-aware refinement** approach that enables direct deployment on real dual-arm hardware; and (3) a **parameterized primitive representation** that supports instance- and category-level transfer with minimal additional data collection. These aspects collectively enable robust, reproducible real-world performance without extensive simulator tuning or long RL training.
>
> Below we will respond to each specific concern raised (on comparative validation vs. RL/simulation, category-level generalization, sensitivity to size estimation, position robustness, and trial counts). We have prepared additional experiments and analyses to address them and will present those results and clarifications in the detailed point-by-point reply.
>
> ***
>
> ## (A) Response to Weakness 1: On the need for comparative evidence supporting the claim that post-optimization is safer and more efficient than RL (Simulation or Real-World)
>
> We thank the reviewer for highlighting the importance of experimental validation. As emphasized throughout the paper, RL-based pipelines remain the dominant paradigm in non-prehensile manipulation, particularly those reliant on simulator-crafted task environments. However, they are inherently **inefficient and non-scalable**, requiring task-specific scene construction (table setup, robot URDF import, object 3D assets), extensive parameter tuning (e.g., mass, friction, elasticity), prolonged exploration, and subsequent sim-to-real adaptation (e.g., DAgger), which inevitably degrades performance and still incurs significant hardware overhead during deployment.
>
> Real-world RL, while bypassing simulation, introduces **substantial operational risks** during early-stage exploration (e.g., object breakage, arm collisions), requiring constant human supervision and intervention, which is an approach rarely applied to challenging non-prehensile tasks.
>
> To provide direct quantitative evidence, we have updated `Table 1` with the average training-time comparison between BiNoMaP and these RL/simulation-based baselines or visuomotor policies (measured after all preconditions are ready). All training was performed on a single RTX 4090 with 24GB memory. The results support our claim that BiNoMaP achieves drastically lower training cost while avoiding the safety and brittleness issues of RL.
>
> ## (B) Response to Weakness 2: On the concern that category-level generalization is limited to size variation rather than shape variation
>
> We appreciate the reviewer's thoughtful concern. Our initial experiments indeed emphasized **size variation within category**, as this setting allows fair and controlled comparison across baselines. However, BiNoMaP is **not limited to size-only generalization**. As shown in the original submission:
>
> > **(1)** The ***pivoting*** skill transfers across *bowl (hemisphere)*, *box (cuboid)*, and *bottle (cylinder)*.
>
> > **(2)** The ***wrapping*** skill transfers across *ball (sphere)* and *basket (cuboid)*.
>
> To further demonstrate **shape-generalization capability**, we performed **additional real-world experiments** in the rebuttal period:
>
> > **(1)** The ***pivoting*** transferred to *big jar* and *big block*, objects with significantly different geometry and mass distribution (including both empty and water-filled jars in various levels).
>
> > **(2)** The ***wrapping*** transferred to the same two new objects (*big jar* and *big block*) beyond ball/basket.
>
> Updated visualizations are provided in `Appendix Fig. 16, 17`, and dynamic rollouts are included in the new supplemental video named `new_exps_jar_block.mp4`. These results confirm that BiNoMaP can rapidly adapt to **novel shapes** with minimal engineering overhead.

---

> ### Author Response · Authors · 2025-11-24
> **Response by Authors to Reviewer AWJr [Part 2]**
>
> ## (C) Response to Weakness 3: On robustness to inaccurate object-size estimation from real-world point clouds
>
> We thank the reviewer for raising this important point. Indeed, real-world perception inevitably introduces noise, unlike the idealized geometric observations available in simulation. BiNoMaP is designed to be **robust by construction**:
>
> > **(1)** Both the dense trajectory extraction and the geometry-aware optimization are performed **directly in the real world**, meaning that optimization is naturally co-adapted to perception noise.
>
> > **(2)** In the iterative adjustment step (`Eq. 4`), the optimized $d_{(\hat{k})}$ typically yields a **slightly over-constrained contact margin**, intentionally tolerating point-cloud estimation errors and preventing arm–object separation, which is a key factor underlying our high success rate.
>
> Only in cases where perception fails catastrophically (e.g., severe mask errors from SAM2 leading to missing or polluted point clouds) does performance significantly degrade. Some failure examples are visualized in `Appendix Fig. 12` (cases `(5), (6), (9), (10)`) and dynamically illustrated in the `supplementary video (03:43–04:16)`. We believe BiNoMaP's robustness to moderate perception noise is a significant practical advantage over sim-to-real methods that depend on noise-free simulated geometry.
>
> ## (D) Response to Weakness 4: On the generalization capability across variations in object position
>
> We agree fully that spatial generalization across object placement is critical. BiNoMaP explicitly incorporates this capability:
>
> > **(1)** As described in the last paragraph of `Sec. 3.2–3.3`, we employ Florence-2 and SAM2 to isolate the object and perform skill-specific adaptation at the **object level**, decoupling execution from global scene layout.
>
> > **(2)** All real-world trials in our evaluation are conducted with randomized object placements, ensuring broad spatial coverage (see illustrations in `Appendix Fig. 10`).
>
> > **(3)** Real robot rollouts in the `supplementary video (03:06–03:42)` show more than 4 distinct placement locations per object, all successfully executed.
>
> To provide more systematic evidence, we additionally conducted a dense grid test on a humanoid dual-arm platform shown in `Fig. 14` for pivoting-bowl:
>
> > **(1)** 22 densely sampled positions across the workspace.
>
> > **(2)** 34 total trials, 22 successes (about 65% success rate).
>
> Visualization is provided in `Appendix Fig. 18`, with the corresponding supplementary video named `new_exps_position_test.mp4`. These results confirm that BiNoMaP maintains strong spatial generalization without retraining.
>
> ## (E) Response to Weakness 5: On the adequacy of the number of evaluation trials
>
> We appreciate the reviewer's attention to statistical reliability. The reviewer's concern appears to refer to the **10-trial evaluation** reported in `Table 1`. In our full experiments (`Tables 2 and 4`), we already use **20 trials** or **50–70 trials**, consistent with standard practice in real-robot manipulation studies.
>
> Nevertheless, we agree that `Table 1` should also reflect a more robust evaluation. We have therefore increased the number of trials **from 10 into 30** for all methods (given the time constraints of the rebuttal period). The updated results remain strongly in favor of BiNoMaP. We thank the reviewer for highlighting this point. The increased trial count strengthens the empirical claims and improves completeness.
>
> ## Concluding Remarks
>
> We sincerely thank the reviewer for the insightful and constructive comments. We have added new experiments (shape generalization, positional robustness), clarified key aspects of the method's design, and expanded `Table 1` with training-time comparisons and increased trial counts. We hope these improvements address the reviewer's concerns and help convey the practicality, generality, and scientific value of BiNoMaP.

---

### Official Review · Reviewer_ESrq · 2025-11-03

**Soundness:** 2
**Presentation:** 2
**Contribution:** 2
**Rating:** 4
**Confidence:** 4

**Summary:**

The paper proposes BiNoMaP, a three-stage, RL-free pipeline for learning bimanual non-prehensile manipulation skills such as poking, pushing, pivoting, and wrapping. The method extracts hand trajectories from human video demonstrations, refines them through a geometry-aware post-optimization procedure, and parameterizes the resulting trajectories by object size and geometry to achieve limited category-level generalization. Experiments show qualitative success on a few simulated dual-arm tasks.

**Strengths:**

1. The proposed approach focuses on bimanual, non-prehensile skills—an under-explored problem setting.
2. Authors propose a RL-free formulation reduces training cost and avoids reward-design issues.
3. The pipeline could be practically useful if shown to generalize well in real-robot experiments.

**Weaknesses:**

**1. Limited Novelty and Contribution to ICLR**

The paper’s technical core is largely procedural, consisting of three sequential steps:
(1) hand-trajectory extraction from videos,
(2) geometric post-optimization, and
(3) VLM-based primitive parameterization for spatial relocation.

While each component is competently executed, the combined framework introduces no fundamentally new algorithmic idea or learning formulation.
The system primarily assembles existing components into a pipeline without proposing new representations, objectives, or architectures for robot learning.
As a result, the contribution feels more like an application note or system integration paper, rather than a conceptual advance suitable for ICLR’s general audience.

Specifically, the motion optimization and “geometry-aware” refinement procedures are handcrafted rather than learned, and the use of VLMs for translation-based relocation (as described near Lines 280–290) represents a straightforward engineering extension, not a novel modeling technique.

I currently do not see a clear learning insight or methodological novelty, which limits its appeal to ICLR’s research-focused readership.

**2. Limited Generalization Capability**

A large portion of the method depends on heuristic, hand-tuned geometric adjustments that raise concerns about robustness and generalization.
For example, the authors describe in Line 278 onward a process termed Geometry-Aware Iterative Contact Adjustment, where contact trajectories are refined through repeated geometric scaling and distance reduction.

This is essentially a manual iterative scaling procedure with several arbitrary hyperparameters (d(1)=5mm, γ=0.85, k≤10), whose stability and physical realism are not theoretically justified.
There is no learning signal or adaptive feedback beyond success/failure checks, meaning the approach cannot generalize beyond objects with similar geometry and contact conditions.

Moreover, the claimed “spatial generalization” at deployment — implemented by translating refined trajectories according to VLM-detected displacements — is limited to planar offsets and does not address variation in object shape, size, or mass distribution. This suggests the method’s generalization is narrowly geometric and likely brittle under real-world variation or perception noise.

**Questions:**

Please see my weaknesses section, thanks!

---

> ### Author Response · Authors · 2025-11-24
> **Response by Authors to Reviewer ESrq [Part 1]**
>
> ## Overall Response
>
> We thank `Reviewer ESrq` for the time and effort spent evaluating our work, as well as for recognizing the importance of studying bimanual non-prehensile manipulation and the practical value of our RL-free formulation. We appreciate these positive remarks and the constructive nature of the feedback.
>
> At the same time, several comments seem to stem from misunderstandings of our core contributions. BiNoMaP is not a procedural assembly, but **a new framework for extracting, refining, and parameterizing bimanual non-prehensile manipulation primitives** from a single human demonstration (**without RL or simulation assets**). Our geometry-aware trajectory refinement and category-level primitive parameterization together enable **generalization across object instances, object categories, and even robot embodiments**, which we believe represents a meaningful methodological advance for the community.
>
> We thank the reviewer for highlighting places where additional clarification is helpful. Below, we address each concern in detail and provide further analysis and experiments to resolve the raised issues.
>
> ***
>
> ## (A) Core Clarification on a Factual Error
>
> We sincerely thank the reviewer for the thoughtful assessment. After carefully reading the review, we identified one factual misunderstanding that is important to clarify: all components of BiNoMaP (including data collection, learning, and evaluation) are **conducted entirely on real dual-arm hardware**, without any simulated environments, simulated tasks, or simulator-specific design choices. No sim-to-real transfer is involved at any stage.
>
> This distinction matters because BiNoMaP is intentionally designed as an **RL-free, sim-free paradigm** that directly addresses the core challenges of real-world bimanual non-prehensile manipulation, such as perception noise, contact uncertainty, and embodiment constraints. By avoiding reward engineering, domain randomization, and simulator-specific task tuning, BiNoMaP does not suffer from the common performance drop seen when transferring RL policies to hardware. Our quantitative (`Tables 1, 4`) and qualitative (`Figs. 3, 4, 11, 15`) studies demonstrate stable performance across object instances, categories, and embodiments. We emphasize this correction to ensure the reviewer's subsequent concerns are interpreted in the proper context.
>
> ## (B) Response to Concern 1: Methodological Novelty and Contribution
>
> First, we appreciate that the reviewer acknowledges the value of pursuing a real-world, RL-free, sim-free paradigm. We elaborate on the novelty of the proposed three-stage framework:
>
> > ### 1. The framework is not a modular assembly of existing components.
> Obtaining usable bimanual non-prehensile trajectories is itself extremely challenging. Through extensive trials using an ALOHA dual-arm robot and a VR-teleoperated humanoid robot, we found that human teleoperation fails to generate stable demonstrations for tasks such as *pivoting a bowl*, often producing discontinuous or noisy motions. We include the evidence in the updated supplementary video named `new_exps_teleop_test.mp4`. This is why prior work overwhelmingly relies on RL in simulators with handcrafted rewards. Our alternative approach extracts non-prehensile manipulation trajectories *directly from human-hand videos*, enabling scalable and low-cost acquisition.
>
> > ### 2. Trajectory extraction and refinement are technically novel and essential.
> Raw hand-pose trajectories contain severe temporal inconsistencies and amplified perception noise. We propose **(1)** a coherent multi-frame hand-pose extraction algorithm (`Algorithm 1`), and **(2)** a coarse-to-fine 3D spatial and rotational refinement procedure (`Sec. 3.3`), which smooths Euclidean trajectories and SO(3) rotations. These refinements are the key enabler for replaying human-like bimanual motions on robots without force or tactile sensing. `Fig. 9` in the appendix visualizes the optimization steps.
>
> > ### 3. Category-level generalization via shape-difference parametrization.
> Unlike RL-based work that binds each skill to a single object instance, BiNoMaP generalizes to large intra-category variability (`Fig. 6`, showing nearly 50 real objects). We compute object-specific size/shape attributes via stereo point clouds and compare them to the demonstration object (`Appendix Fig. 10`), enabling efficient instance- and category-level transfer.
>
> > ### 4. Extensive hardware evaluations validate practicality.
> Videos in the supplementary material show high-quality rollouts and smooth trajectories across diverse scenarios. These results indicate that our paradigm provides a credible alternative to heavy simulator–RL pipelines.
>
> We respectfully suggest that the reviewer reconsider the contribution in light of these clarifications. Introducing a robust, scalable, and real-world–oriented paradigm for non-prehensile manipulation is, we believe, a meaningful step toward practical embodied intelligence.

---

> ### Author Response · Authors · 2025-11-24
> **Response by Authors to Reviewer ESrq [Part 2]**
>
> ## (C) Response to Concern 2: Limited Generalization Capability
>
> We thank the reviewer for raising this important point and respond as follows:
>
> > ### 1. The optimization stage is lightweight and performed only once per skill.
> Although BiNoMaP performs a single-stage trajectory smoothing and contact-consistent adjustment, this process does not require RL-based reward design or neural network training. The same hyperparameters apply across all tasks and all object instances. Once optimized, the primitives generalize **(i)** spatially to different object poses of the same instance, and **(ii)** across object categories with varied shapes, sizes, and mass distributions (`Tables 2–3`). The real-world cost remains significantly lower than RL or imitation-learning pipelines.
>
> > ### 2. Generalization is extensively validated in the original submission.
> BiNoMaP successfully transfers skills across objects with substantial variation in shape, appearance, or mass (`Table 4` and `Figs. 3, 6, 11`). Videos further confirm robust real-world behavior.
>
> > ### 3. We conducted additional experiments for more reliable evidence.
> To further demonstrate generalization, we added challenging new transfers: (1) **Pivoting** → large jars, large blocks, and jars with different water fill levels (mass variation). (2) **Wrapping** → jars and blocks beyond the original balls and baskets. Updated visualizations are provided in `Appendix Fig. 16, 17`, and videos are included in the newly uploaded supplementary material with the name `new_exps_jar_block.mp4`. We believe that these new experiments further address concerns about the BiNoMaP's generalization when there has variation in object shape, size, or mass distribution.
>
> These new rollouts provide strong additional evidence that BiNoMaP supports efficient and reliable transfer to unseen objects with substantial geometric and physical differences.
>
> ## Concluding Remarks
>
> We deeply appreciate the reviewer's time, constructive feedback, and engagement with our work. We hope the above clarifications address the concerns raised, especially concerning the real-world, RL-free, sim-free nature of the framework, the technical novelty of our trajectory extraction and refinement pipeline, and the extensive new evidence of generalization. We believe BiNoMaP offers a principled and practically meaningful alternative for real-world non-prehensile manipulation, and we respectfully ask the reviewer to reconsider the contribution in this light.

---

### Author Response · Authors · 2025-11-24
**Author Final Remarks**

We sincerely thank all reviewers for their constructive and insightful feedback. Their comments have significantly improved both the clarity and completeness of the manuscript. In our revised submission, we carefully addressed every question and concern across all three reviews and further strengthened the work with new analyses, expanded experiments, and clearer methodological descriptions. Below, we summarize our major revisions and responses.

***

## 1. Clarifying methodological assumptions and terminology.
Reviewers raised concerns regarding the use of "contact-rich" to describe non-prehensile skills and the geometric assumptions used in our trajectory formulation. We clarified that our terminology follows long-standing conventions in the manipulation community, where non-prehensile actions require continuous contact analogous to "clamped phases" in prehensile tasks. We also explained in detail why the coplanarity assumption applies only to end-effector positional DOFs and does not restrict rotational DOFs, and why it remains widely valid across all four skills and all object categories. Several typographical issues and notational inconsistencies (e.g., previous use of *\texttt*) were corrected as well.

## 2. Improving related-work coverage and avoiding overclaims.
We followed the reviewers' suggestions to integrate missing but highly relevant literature (including modular motion primitives and recent bimanual large-behavioral-models), and removed a potentially overclaimed sentence from the introduction. These updates make clearer what our contributions are, how BiNoMaP differs from prior art, and why our problem formulation is both well-motivated and practically relevant.

## 3. Enhanced explanation of human-hand demonstrations.
In response to questions regarding the necessity and procedure of human demonstration capture, we added a detailed paragraph clarifying why human bimanual motions are essential for initializing complex non-prehensile trajectories, the constraints needed to ensure stable perception and consistent hand–object contact, and why more anthropomorphic or free-form demonstration styles would introduce ambiguity detrimental to trajectory extraction and subsequent optimization.

## 4. Fair and rigorous reproduction of all six baselines.
We substantially expanded the description of how we reproduced the three visuomotor policy baselines (ACT, DP, DP3) and the three RL-based sim-to-real methods (HACMan, CORN, DyWA). We explained the decisions made to ensure fairness—such as using optimized BiNoMaP executions to obtain clean training data, adhering strictly to each method's architecture and hyperparameters, recreating simulation environments faithfully, and disabling additional stabilizers during deployment. We also summarized the structural reasons why these baselines inherently struggle with dense-contact, dual-arm tasks, clarifying why their performance in `Tab. 1` and `Tab. 4` is significantly lower than BiNoMaP despite careful reproduction.

## 5. Substantial new experiments demonstrating generalization and adaptability.
To further address concerns about generalization, we added an entirely new `Supplementary Section G`: **Further Verification of Generalization and Adaptability**, which includes: (1) **New object-shape generalization experiments**, extending pivoting to big jar and big block (including variations in mass distribution), and extending wrapping to two additional object shapes. These results demonstrate rapid and robust transfer of primitives to previously unseen geometries. (2) **Dense spatial generalization tests**, using a humanoid dual-arm robot to perform pivoting on 30+ distinct bowl placements, achieving ~22/34 success.

Corresponding visualizations (`Figs. 16–17`) and new videos have been added to the Supplement (`new_exps_teleop_test.mp4`, `new_exps_jar_block.mp4`, and `new_exps_position_test.mp4`).

## 6. Additional clarifications and corrections.
We clarified the role of one-time optimization of contact distances, the robot's behavior on materials with extreme rigidity, the limitations on objects requiring deformation-free contact, and the precise use of VLM-based segmentation during policy deployment.

***

In sum, the revised manuscript incorporates extensive clarifications, additional ablations, an entirely new generalization section, improved baseline analysis, expanded related work, and corrected technical details. We hope these revisions adequately address all reviewer concerns and demonstrate the robustness, practicality, and novelty of the proposed BiNoMaP framework. We thank the reviewers again for their thoughtful comments and valuable guidance.

---

### Meta-Review · Area_Chair_xScQ · 2026-01-08

**Summary:**

This paper proposes BiNoMaP, an RL-free and sim-free pipeline for acquiring bimanual non-prehensile manipulation primitives (poking, pushing, pivoting, wrapping) from a single human-hand video demonstration, followed by geometry-aware trajectory refinement and parameterized transfer across object instances and robot embodiments. The topic is timely and practically relevant: robust dual-arm contact-intensive manipulation remains difficult for RL and large visuomotor policies, especially when real-world data collection is costly and sim-to-real gaps are large.

The main remaining concern is that some reviewers still perceive the contribution as a procedural integration rather than a broadly general methodological advance for an ML audience. Some geometric assumptions made in the optimization process appear to work for the selected primitives and objects.

The authors should also consider a comparison with other geometric-based or contact-based skill transfer, such as "You Only Demonstrate Once: Category-Level Manipulation from Single Visual Demonstration" and "Physics-Driven Data Generation for Contact-Rich Manipulation via Trajectory Optimization."

**Reviewer Concerns:**

ESrq:
Addressed: factual misunderstanding about simulation, generalization beyond simple planar translation, heuristic contact adjustment.
Outstanding: Core novelty in an ML context, theoretical/algorithmic justification of the iterative geometric procedure.

AWJr:
Addressed: need for comparative evidence, shape-level generalization (not just size), robustness to object position variation, statistical reliability
Outstanding: robustness to inaccurate size estimation, generality of the pipeline in more diverse contact conditions

VeaC:
Addressed: “contact-rich” terminology, overclaim about "first RL-free," missing related work, coplanarity assumption, human demonstrations being "unnatural", baseline reproduction
Outstanding: Physical property generalization

**Reviewer Scores:**

Reviewer ESrq is unlikely to increase their score, as the key concern regarding novelty in the machine learning context remains unaddressed.
Reviewer AWJr may slightly increase their score, but would likely still have concerns about shape-level generalization and the generality of the pipeline under more diverse contact conditions.
Reviewer VeaC would likely maintain their current score (accept).

Overall, this remains a borderline paper (ESrq: reject, AWJr: borderline, VeaC: accept). I would vote to reject the paper, as it does not sufficiently address novelty relative to prior robotics methods and offers limited contributions that are meaningful to the broader machine learning community.

---

### Decision · Program_Chairs · 2026-01-26

Reject